# ACLY ubiquitination by CUL3-KLHL25 induces the reprogramming of fatty acid metabolism to facilitate iTreg differentiation

**Miaomiao Tian[1†], Fengqi Hao[1†], Xin Jin[1], Xue Sun[1], Ying Jiang[1], Yang Wang[1], Dan Li[2,3], Tianyi Chang[1], Yingying Zou[1], Pinghui Peng[1], Chaoyi Xia[1], Jia Liu[1], Yuanxi Li[1], Ping Wang[4], Yunpeng Feng[1]\*, Min Wei[1]\***

[1]Key Laboratory of Molecular Epigenetics of the Ministry of Education (MOE), Northeast Normal University, Changchun, China; [2]Shanghai Institute of Immunology, Shanghai Jiao Tong University School of Medicine, Shanghai, China; [3]Department of Immunology and Microbiology, Shanghai Jiao Tong University School of Medicine, Shanghai, China; [4]Tongji University Cancer Center, Shanghai Tenth People's Hospital, School of Medicine, Tongji University, Shanghai, China

**\*For correspondence:**
fengyp0108@nenu.edu.cn (YF);
weim750@nenu.edu.cn (MW)

[†]These authors contributed equally to this work

**Competing interests:** The authors declare that no competing interests exist.

**Abstract** Inducible regulatory T (iTreg) cells play a central role in immune suppression. As iTreg cells are differentiated from activated T (Th0) cells, cell metabolism undergoes dramatic changes, including a shift from fatty acid synthesis (FAS) to fatty acid oxidation (FAO). Although the reprogramming in fatty acid metabolism is critical, the mechanism regulating this process during iTreg differentiation is still unclear. Here we have revealed that the enzymatic activity of ATP-citrate lyase (ACLY) declined significantly during iTreg differentiation upon transforming growth factor β1 (TGFβ1) stimulation. This reduction was due to CUL3-KLHL25-mediated ACLY ubiquitination and degradation. As a consequence, malonyl-CoA, a metabolic intermediate in FAS that is capable of inhibiting the rate-limiting enzyme in FAO, carnitine palmitoyltransferase 1 (CPT1), was decreased. Therefore, ACLY ubiquitination and degradation facilitate FAO and thereby iTreg differentiation. Together, we suggest TGFβ1-CUL3-KLHL25-ACLY axis as an important means regulating iTreg differentiation and bring insights into the maintenance of immune homeostasis for the prevention of immune diseases.

## Introduction

As a critical subset of immunosuppressive cells, regulatory T (Treg) cells play a central role in the suppression of aberrant or excessive immune responses and are critical for the maintenance of immune homeostasis (*Dominguez-Villar and Hafler, 2018*). Defects in Treg frequency or function are often associated with distinct autoimmune disorders, such as system lupus erythematosus (SLE), rheumatoid arthritis (RA), and inflammatory bowel disease (IBD) (*Sakaguchi et al., 2008*; *Tao et al., 2017*). Treg cells are composed of thymus-derived natural Treg (nTreg) and peripheral inducible Treg (iTreg) cells, which are originated and matured at different locations in the body (*Lee et al., 2009*; *Noack and Miossec, 2014*). In general, nTreg cells are dominant in the periphery under unperturbed conditions, whereas iTreg cells are often boosted upon special immune challenges to balance inflammatory responses (*Murai et al., 2010*). Furthermore, enhancing iTreg cells by stimulating iTreg differentiation or employing adoptive transfer appeared to be effective in the therapeutic treatment for several immune diseases (*Dall'Era et al., 2019*; *Esensten et al., 2018*; *Göschl et al., 2019*; *Ryba-Stanisławowska et al., 2019*).

iTreg cells are derived from activated T (Th0) cells, which can also differentiate into effector CD4$^+$ T cells (Teffs), including Th1, Th2, and Th17, depending on cytokine contexts, and the establishment of iTreg cells relies on transforming growth factor β1 (TGFβ1) (*Chen et al., 2003*; *Maciolek et al., 2014*). It has been characterized that TGFβ1 stimulates the phosphorylation of transcription factors Smad2 and Smad3 to facilitate the formation of a heterotrimeric complex containing Smad2/3/4 (*Derynck and Zhang, 2003*; *Souchelnytskyi et al., 1997*). Subsequently, this Smad2/3/4 complex binds to the conservative non-coding sequence 1 (CNS1) in *Foxp3,* which encodes the master transcription factor critical for the establishment and maintenance of iTreg cells, thereby promoting iTreg differentiation (*Chen et al., 2011*; *Fontenot et al., 2003*; *Hori et al., 2003*; *Schlenner et al., 2012*).

Accumulating evidence has revealed that different types of T cells are usually associated with distinct metabolic characteristics (*Chen et al., 2015*; *Kempkes et al., 2019*). For instance, Th0, Th1, Th2, and Th17 prefer de novo fatty acid synthesis (FAS) that substantially supports anabolism to meet the need for biological macromolecules during rapid proliferation (*Lochner et al., 2015*; *Ma et al., 2017*; *Maciolek et al., 2014*; *Wang et al., 2011*). In contrast, iTreg relies on fatty acid oxidation (FAO) (*Chen et al., 2015*; *Gerriets et al., 2015*; *Ma et al., 2017*; *Michalek et al., 2011*). Because carnitine palmitoyltransferase 1 (CPT1) is the rate-limiting enzyme in FAO, upregulating CPT1 by its substrate palmitate and downregulating CPT1 by the specific inhibitor etomoxir, respectively, could result in enhancement and impairment in FAO and iTreg differentiation (*Gualdoni et al., 2016*; *Michalek et al., 2011*). Obviously, FAS and FAO are reciprocal pathways, only one of which can be dominant in a specific type of T cells (*Foster, 2012*; *Wolfgang and Lane, 2006*). During iTreg differentiation, cell metabolism undergoes a series of dramatic changes, including a shift from FAS to FAO. Although the downregulation of FAS appears to be important (*Berod et al., 2014*), the mechanism regulating FAS in the process of iTreg differentiation is still unclear.

FAS is directly controlled by a series of metabolic enzymes in cytoplasm, including three rate-limiting enzymes ATP-citrate lyase (ACLY), acetyl-CoA carboxylase (ACC), and fatty acid synthase (FASN) (*Lochner et al., 2015*). In brief, ACLY converts mitochondrial-derived citrate into acetyl-CoA and oxaloacetic acid (OAA), and provides the main acetyl-CoA source for de novo FAS. ACC catalyzes acetyl-CoA into malonyl-CoA to provide an active two-carbon-unit donor for carbon chain extension. FASN uses both malonyl-CoA and acetyl-CoA as substrates to continuously synthesize long-chain fatty acids. Although regulation of the three enzymes at the transcriptional level was reported (*Kidani et al., 2013*), direct modulation of their activities by posttranslational modifications (PTMs), particularly ubiquitination that is tightly coupled with protein degradation, has set a new stage for the exploration of FAS control. For instance, nutrient deficiency induced ACLY ubiquitination at K540/K546/K554 in human lung cancer cells. This change resulted in the degradation of ACLY proteins and suppressed de novo lipid synthesis, eventually causing impairment in cell proliferation (*Lin et al., 2013*; *Zhang et al., 2016*). ACC stability as well as de novo lipid synthesis in mouse adipocytes was found to be regulated by ubiquitination (*Qi et al., 2006*). Additionally, FASN ubiquitination in HEK293T cells was also linked to its degradation, which in turn compromised de novo lipid synthesis (*Lin et al., 2016*). However, it still remains poorly understood whether and how ubiquitination is involved in the control of FAS during iTreg differentiation.

In this study, we have revealed that ACLY activity declined significantly as iTreg cells were differentiated. Upregulation in ACLY expression hindered the switch from FAS to FAO and thereby impaired iTreg differentiation. ACLY-dependent malonyl-CoA, an intermediate in FAS, could directly finetune CPT1 activity and therefore FAO. In response to TGFβ1 stimulation, CUL3-KLHL25 mediated ACLY ubiquitination to facilitate its degradation. When we removed CUL3 to block ACLY degradation, iTreg differentiation and colitis alleviation by adoptive iTreg cells in mice were compromised. Simultaneous depletion of ACLY partially rescued CUL3 deficiency-induced defects in iTreg differentiation and colitis alleviation. Overall, we demonstrated the regulation of iTreg differentiation via TGFβ1-CUL3-KLHL25-ACLY axis and bring insights into the maintenance of immune homeostasis for the prevention of colitis.

## Results

### ACLY is critical for the efficient differentiation of iTreg cells

To understand the alteration in FAS during iTreg differentiation, we examined the three rate-limiting enzymes ACLY, ACC, and FASN in FAS pathway. It appeared that the enzymatic activity of ACLY was reduced markedly, while ACC and FASN activities remained largely unchanged (*Figure 1A*, *Figure 1—figure supplement 1A*). Meanwhile, acetyl-CoA and OAA, the two products of ACLY-mediated reaction, were also reduced upon TGFβ1 treatment, confirming the decreased ACLY activity during iTreg differentiation (*Figure 1B*, *Figure 1—figure supplement 1B*). Importantly, preventing the decline in ACLY activity by supplementing cells with exogenous ACLY drastically impaired iTreg differentiation (*Figure 1C*, *Figure 1—figure supplement 1C*). In contrast, knocking down *Acly* by small interfering RNAs (siRNAs) or blocking ACLY activity by the specific inhibitor SB204990 led to elevated iTreg differentiation (*Figure 1D–E*, *Figure 1—figure supplement 1D-E*). Together, these data suggest that ACLY regulates iTreg differentiation.

### ACLY is required for the reprogramming of fatty acid metabolism during iTreg differentiation

ACLY converts citrate into acetyl-CoA and OAA, and acetyl-CoA is the main source supporting de novo FAS, mevalonate-cholesterol synthesis, and histone acetylation (*Zaidi et al., 2012*). To further

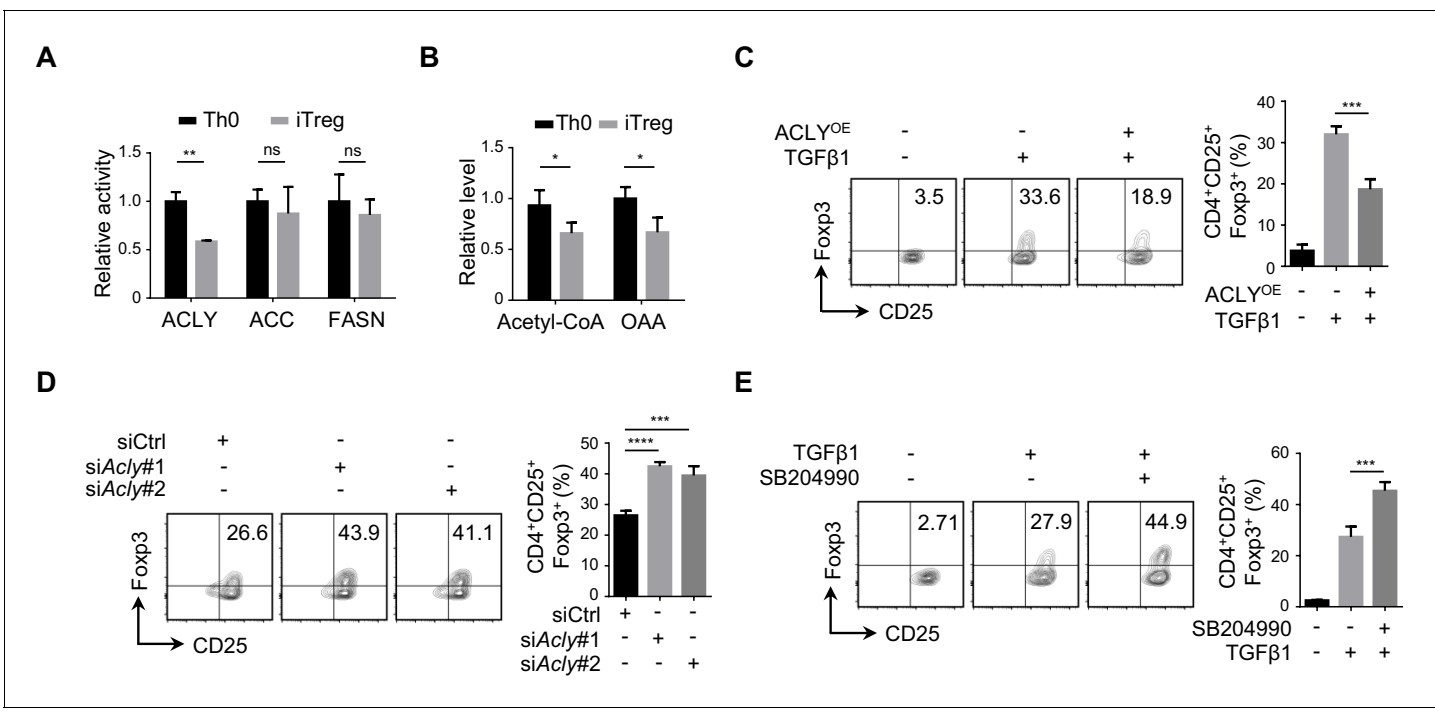

**Figure 1.** ACLY is critical for iTreg cell differentiation. (A, B) Detection of ATP-citrate lyase (ACLY) activity in inducible regulatory T (iTreg) cells. Naive CD4+ T cells isolated from mice were cultured with Dynabeads Mouse T-Activator CD3/CD28 and rmIL-2 for 72 hr to obtain activated T (Th0) cells or simultaneously supplemented with rhTGFβ1 (2 ng/ml) to generate iTreg cells. (A) Assessment of enzymatic activity for ACLY, acetyl-CoA carboxylase (ACC), and fatty acid synthase (FASN) from Th0 or iTreg cells. (B) Assay for cytosol acetyl-CoA and oxaloacetic acid (OAA) from Th0 or iTreg cells. (C) ACLY overexpression affects iTreg differentiation. Naive CD4+ T cells were transfected with GFP-ACLY and cultured under Th0- or iTreg-polarization condition as in (A). Cells were analyzed by flow cytometry (FCM) to evaluate GFP+CD25+Foxp3+ iTreg cells. (D, E) Inhibition of ACLY influences iTreg differentiation. Naive CD4+ T cells transfected with small interfering RNAs (siRNAs) against *Acly* (D) or treated with SB204990 (100 μM) (E) were cultured under Th0- or iTreg (with 0.5 ng/ml rhTGFβ1)-polarization condition for 72 hr. CD4+CD25+Foxp3+ iTreg cells were assayed by FCM (left) and quantified (right). Data represent mean ± SD of three independent experiments, with significance determined by Student's t-test (A, B) or one-way analysis of variance (ANOVA) test (C–E). *p<0.05, **p<0.01, ***p<0.001, and ****p<0.0001; ns, nonsignificant.

The online version of this article includes the following figure supplement(s) for figure 1:

**Figure supplement 1.** Analyses of ACLY activity during iTreg differentiation.

understand ACLY-dependent regulation of metabolic reprogramming during iTreg differentiation, we performed [U-$^{13}$C] glucose-tracing experiment. Upon ACLY inhibition, $^{13}$C-labeled fatty acids were reduced as expected (*Figure 2A*). However, key intermediates in mevalonate-cholesterol synthesis, including 3-hydroxy-3-methyl glutaryl coenzyme A (HMG-CoA), mevalonate, mevalonate 5-pyrophosphate, and cholesterol remained rather stable irrespective of the presence of ACLY inhibitor (*Figure 2—figure supplement 1A–D*). Exogenous supplement of mevalonate and cholesterol failed to affect ACLY-inhibition-induced enhancement in iTreg differentiation (*Figure 2—figure supplement 2A–C*), confirming a dispensable role of mevalonate-cholesterol synthesis in ACLY-dependent regulation of iTreg differentiation. Meanwhile, we examined the level of nuclear acetyl-CoA that directly supports histone acetylation and found that it remained relatively stable, regardless of the presence of ACLY inhibitor (*Figure 2—figure supplement 3A–B*). In line with this observation, acetylation in total histones and H3/H4 was not influenced by ACLY inhibition (*Figure 2—figure supplement 3C–D*).

Intriguingly, adding back palmitate did not change ACLY-inhibition-induced iTreg differentiation (*Figure 2B*, *Figure 2—figure supplement 4A*), despite a significant reduction in FAS upon ACLY inhibition (*Figure 2A*). Considering that palmitate, the terminal product of FAS, may not be the limiting factor coordinating ACLY-dependent impact and iTreg differentiation, other intermediate products in FAS can be potentially at play (*Figure 2—figure supplement 4B*). Next, we targeted key enzymes in FAS separately with different inhibitors to gain insights into the pivotal metabolite(s). While FASN inhibition abolished ACLY-inhibition-induced iTreg differentiation, the inhibition of ACC barely had any effect (*Figure 2—figure supplement 4C*). This strongly indicates that malonyl-CoA, the substrate of FASN, might be important for ACLY-inhibition-induced iTreg differentiation. Upon ACLY inhibition, we did observe a clear reduction in malony-CoA (*Figure 2—figure supplement 4D*). More importantly, add-back of malonyl-CoA completely abolished ACLY-inhibition-induced iTreg differentiation (*Figure 2C*). These data suggest that ACLY-dependent regulation of iTreg differentiation is probably through malonyl-CoA.

Because malonyl-CoA is a well-characterized metabolic intermediate capable of inhibiting CPT1, the key enzyme in FAO, we asked whether ACLY inhibition could regulate CPT1 activity and thereby FAO to promote iTreg differentiation (*Figure 2—figure supplement 4B*). When ACLY was inhibited by SB204990, both CPT1 activity (*Figure 2D*) and FAO (*Figure 2E*) were enhanced. Under the same condition, upregulating the cellular level of malonyl-CoA by exogenous supplementation of malonyl-CoA or FASN inhibition abrogated ACLY-inhibition-induced upregulation in CPT1 activity (*Figure 2D*, *Figure 2—figure supplement 5A*) and FAO (*Figure 2E*). Importantly, knockdown or inhibition of CPT1 abolished ACLY-inhibition-induced iTreg differentiation (*Figure 2F*, *Figure 2—figure supplement 5B*). Collectively, the reduction in malonyl-CoA resulted from ACLY inhibition mediates the derepression of CPT1 to facilitate FAO and iTreg differentiation.

## TGFβ1 induces ACLY ubiquitination and degradation during iTreg differentiation

To further understand the decline in ACLY activity during iTreg differentiation, we first isolated ACLY proteins from Th0 and iTreg cells and normalized them to the same level (*Figure 3—figure supplement 1A–B*). Interestingly, both ACLY activity and Ser455 phosphorylation, which is a marker directly reflecting the enzymatic activity of ACLY (*Das et al., 2015*; *Sivanand et al., 2017*), remained unchanged (*Figure 3—figure supplement 1A–B*). However, a dramatic decrease in ACLY protein, but not in its mRNA, was observed in the process of iTreg differentiation (*Figure 3A*, *Figure 3—figure supplement 1C*), strongly arguing a mechanism regulating ACLY at its protein level.

Given that protein levels are controlled by both synthesis and degradation, we next examined ACLY protein level upon the treatment of cycloheximide (CHX), a well-documented ribosome inhibitor, often used for blocking protein synthesis (*Schneider-Poetsch et al., 2010*). Our time-course analysis showed that TGFβ1 stimulation caused a reduction in ACLY protein even in the presence of CHX (*Figure 3B*), indicating that protein degradation played an important role in TGFβ1-mediated ACLY downregulation. Indeed, when we blocked protein degradation with MG132, which inhibits ubiquitination-proteasome degradation (*Liu et al., 2016*), TGFβ1-induced downregulation in ACLY was abolished (*Figure 3C*). In contrast, the addition of leupeptin, which blocks autophagy-lysosome degradation (*Liu et al., 2016*), did not affect ACLY protein level (*Figure 3C*). These results

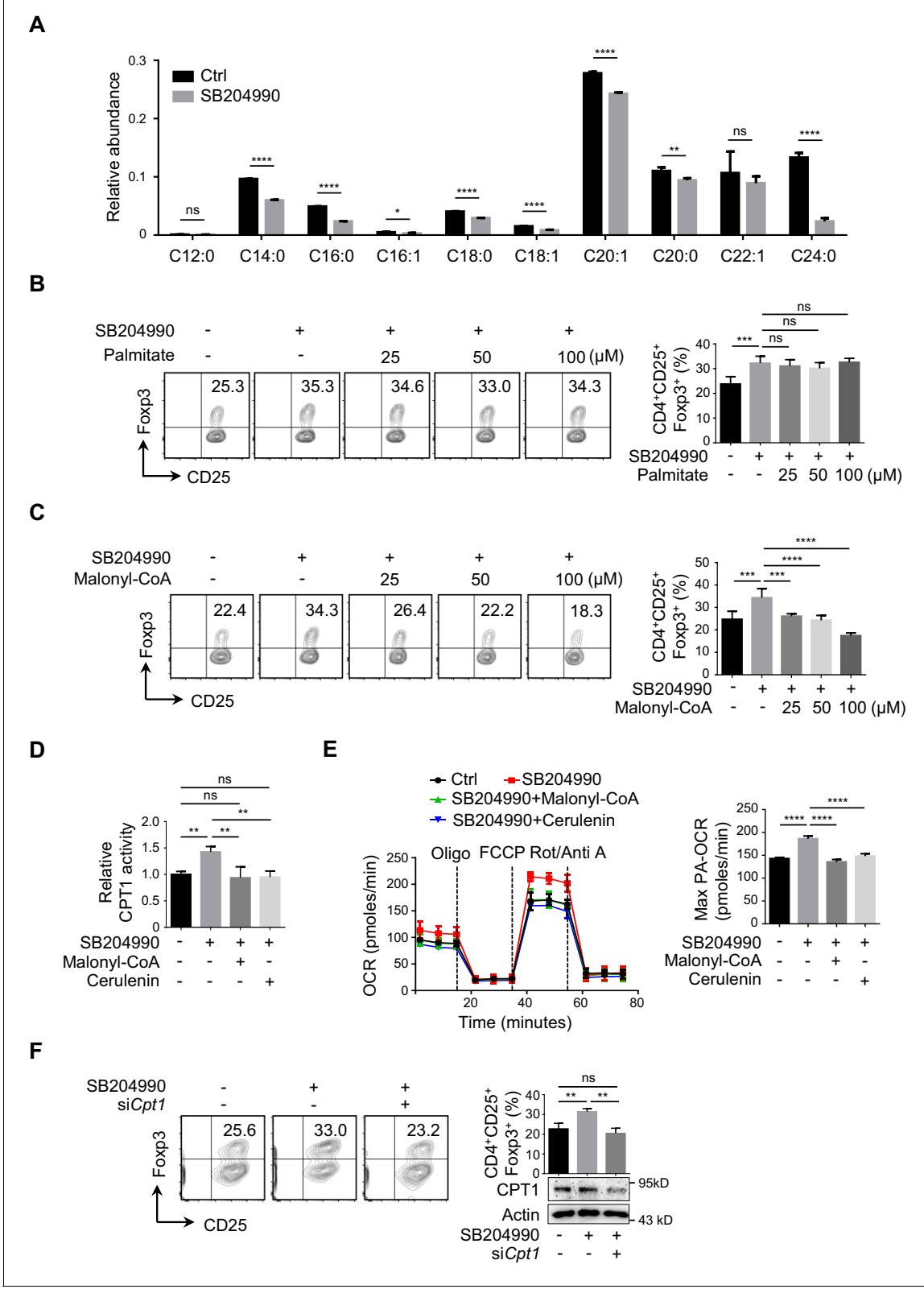

**Figure 2.** ACLY inhibition induces the reprogramming in fatty acid metabolism during iTreg differentiation. (**A**) ATP-citrate lyase (ACLY) inhibition reduces de novo fatty acid synthesis (FAS). Naive CD4+ T cells were treated with SB204990 (100 μM) and cultured under inducible regulatory T (iTreg)-polarization condition as in **Figure 1E** for 24 hr in the presence of [U-13C] glucose (11 mM). Cells were collected and subjected to metabolic flux analysis for FAS by ultra-high performance liquid chromatography-high resolution mass spectrometry (UHPLC-HRMS) analysis. n = 4, Student's t-test,

*Figure 2 continued on next page*

*Figure 2 continued*

mean ± SD. *p<0.05, **p<0.01, ****p<0.0001; ns, nonsignificant. (B, C) Functional evaluation of metabolic intermediates from FAS on iTreg differentiation. Naive CD4$^+$ T cells were treated with SB204990 (100 µM) and cultured as in *Figure 1E* in the presence of different doses of palmitate (B) or malonyl-CoA (C). CD4$^+$CD25$^+$Foxp3$^+$ iTreg cells were assayed by flow cytometry (FCM) (left) and quantified (right). (D, E) ACLY inhibition increases carnitine palmitoyltransferase 1 (CPT1) activity and fatty acid oxidation (FAO). In the presence of malonyl-CoA (50 µM) or cerulenin (4 µM), naive CD4$^+$ T cells treated with SB204990 were cultured as in *Figure 1E* for 24 hr. Cell lysates were used for analyzing CPT1 activity (D). For oxygen consumption rate (OCR) detection, cells were transferred to XF Base Medium containing palmitate and carnitine. Diagram (left) illustrating the OCR at various conditions and associated quantifications (right) are shown (E). (F) Impact of *Cpt1* knockdown on iTreg differentiation. Naive CD4$^+$ T cells transfected with small interfering RNA (siRNA) against *Cpt1* were cultured as in *Figure 1E*. CD4$^+$CD25$^+$Foxp3$^+$ iTreg cells were assayed by FCM (left) and quantified (right). (B–F) Data represent mean ± SD of three (D–F) or four (B, C) independent experiments, with significance determined by one-way analysis of variance (ANOVA) test. **p<0.01, ***p<0.001, and ****p<0.0001; ns, nonsignificant.

The online version of this article includes the following figure supplement(s) for figure 2:

**Figure supplement 1.** Detection of metabolic intermediates in mevalonate-cholesterol synthesis pathway upon ACLY inhibition.

**Figure supplement 2.** Impact of mevalonate and cholesterol on iTreg differentiation.

**Figure supplement 3.** Examination of nuclear acetyl-CoA and histone acetylation upon ACLY inhibition.

**Figure supplement 4.** Functional comparison between ACC and FASN in ACLY-inhibition-induced iTreg differentiation.

**Figure supplement 5.** Functional evaluation of CPT1 in ACLY-inhibition-induced iTreg differentiation.

suggested that, during iTreg differentiation, TGFβ1 induced ACLY downregulation via ubiquitination-proteasome degradation.

Subsequently, we carried on confirming whether ubiquitination determined ACLY stability during iTreg differentiation. In response to TGFβ1 stimulation, ACLY ubiquitination was evidently increased (*Figure 3D–E*). Mutation of ACLY at K530, K536, and K544 (3KR), which are homologous to the three ubiquitination sites (K540, K546, and K554) reported in human ACLY (*Figure 3—figure supplement 2A*; *Lin et al., 2013*; *Zhang et al., 2016*), abolished TGFβ1-dependent upregulation in ACLY ubiquitination (*Figure 3F*). Importantly, the reduction of ACLY stimulated by TGFβ1 was blocked when K530, K536, and K544 were mutated (*Figure 3G*, *Figure 3—figure supplement 2B*). As a consequence, TGFβ1-induced increase in CPT1 activity and iTreg differentiation were diminished (*Figure 3H–I*). Together, TGFβ1 promotes ACLY ubiquitination at K530, K536, and K544 to induce ACLY degradation, which in turn facilitates iTreg differentiation.

## CUL3-KLHL25 is responsible for TGFβ1-mediated ACLY ubiquitination

In human cells, ACLY ubiquitination was found to be regulated by ubiquitin ligases UBR4 and CUL3-KLHL25 and deubiquitinase USP13 (*Han et al., 2016*; *Lin et al., 2013*; *Zhang et al., 2016*). To substantiate the determinants for ACLY ubiquitination in iTreg cells, we examined which of these candidate regulators interacted with ACLY. CUL3-KLHL25 and USP13, but not UBR4, appeared to interact with ACLY in iTreg cells (*Figure 4—figure supplement 1A–B*). Importantly, co-immunoprecipitation demonstrated that CUL3-KLHL25 association, rather than USP13 interaction, with ACLY was dramatically changed by TGFβ1 (*Figure 4A*, *Figure 4—figure supplement 1C*). This strongly suggested an important role for CUL3-KLHL25 in TGFβ1-dependent ACLY ubiquitination during iTreg differentiation.

It is known that CUL3-KLHL25 complex belongs to Cullin–RING ubiquitin ligase family, the largest class of ubiquitin ligases (*Chen and Chen, 2016*; *Zheng and Shabek, 2017*). In brief, CUL3 is the core scaffolding protein holding the entire complex together, while KLHL25 serves as an adaptor for substrate recognition (*Zhang et al., 2016*). When we knocked down *Cul3* or *Klhl25* with targeting siRNAs, ACLY ubiquitination was dramatically reduced (*Figure 4B*, *Figure 4—figure supplement 2A*). Also, both ACLY ubiquitination and degradation induced by TGFβ1 were blocked upon the depletion of CUL3 or KLHL25 (*Figure 4B–C*, *Figure 4—figure supplement 2B*). Moreover, the removal of CUL3 also resulted in an upregulated malonyl-CoA level (*Figure 4—figure supplement 2C*), reduced CPT1 activity, and downregulated iTreg differentiation (*Figure 4D–E*). Importantly, ACLY depletion partially rescued *Cul3*-knockdown-induced alterations in malonyl-CoA level, CPT1 activity, and iTreg differentiation (*Figure 4—figure supplement 2C*, *Figure 4D–E*). These data demonstrated that TGFβ1-mediated ACLY ubiquitination and degradation rely on CUL3-KLHL25.

In addition, we generated CD4$^+$ T-cell-specific CUL3-deficient mice (*Cd4$^{Cre}$Cul3$^{fl/fl}$*) by crossing *Cd4$^{Cre}$* mice (*Dong et al., 2008*) with *Cul3$^{fl/fl}$* mice (*McEvoy et al., 2007*; *Figure 4—figure*

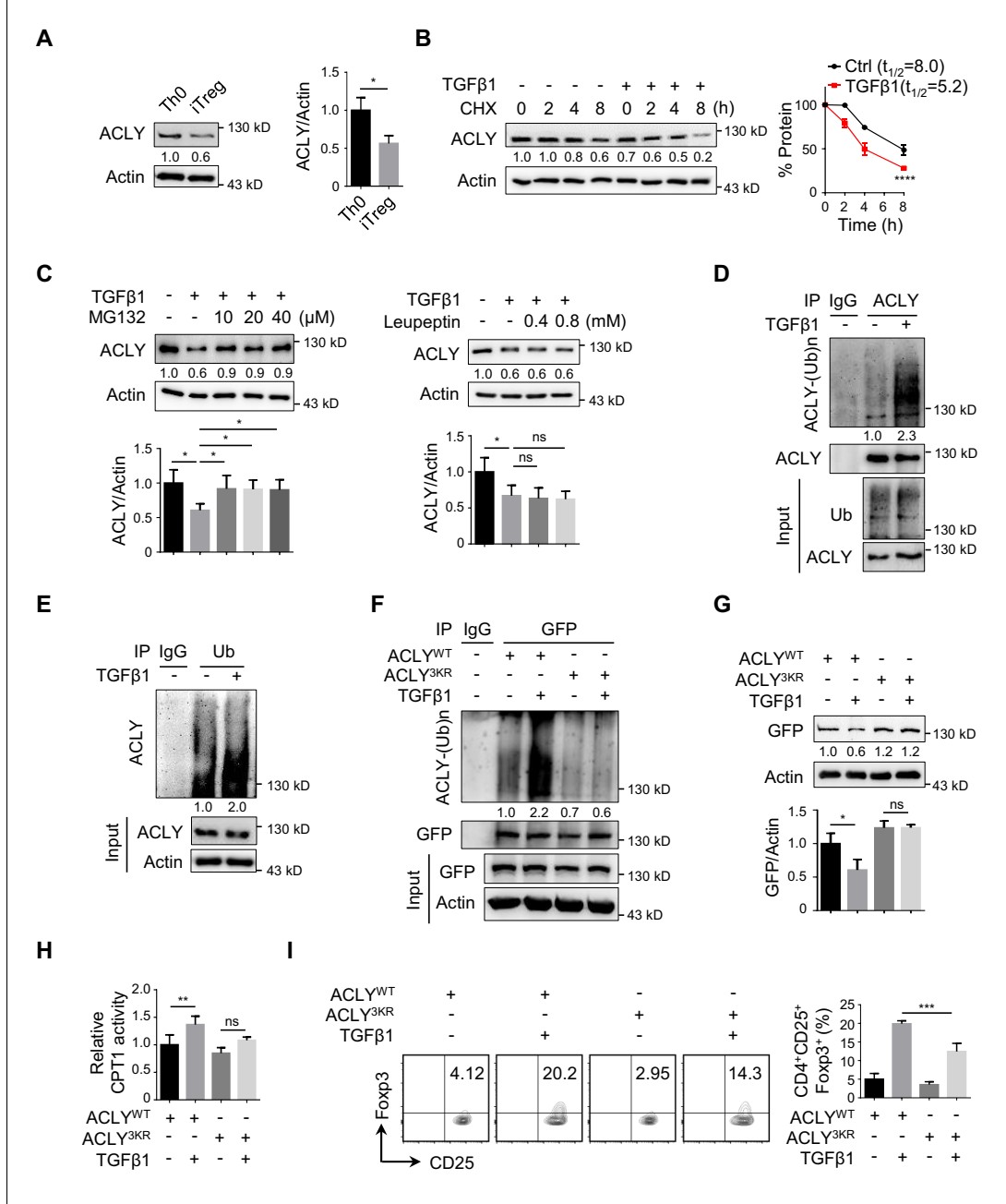

**Figure 3.** TGFβ1 induces ACLY ubiquitination and degradation during iTreg differentiation. (**A**) Assay of ATP-citrate lyase (ACLY) protein level. Activated T (Th0) and inducible regulatory T (iTreg) cells prepared as in *Figure 1A* were subjected to western blot (WB) with antibodies against ACLY (left). Quantification of ACLY levels (right). (**B**) Impact of transforming growth factor β1 (TGFβ1) on ACLY degradation. Naive CD4+ T cells were cultured as in *Figure 1A* for 24 hr and treated with cycloheximide (CHX) (10 μg/ml) as indicated prior to WB analysis (left). Quantification of ACLY levels (right). (**C**) Pathway analysis for ACLY degradation. Th0 and iTreg cells polarized as in *Figure 1A* for 24 hr were treated with MG132 (left) or leupeptin (right) and analyzed with WB for ACLY protein. (**D, E**) Impact of TGFβ1 on ACLY ubiquitination. Cells polarized as in *Figure 1A* for 24 hr were treated with MG132 (10 μM) and cell extracts were immunoprecipitated with antibodies against ACLY (**D**) or ubiquitin (Ub) (**E**). IgG serves as a negative control. (**F–H**) Functional comparison between ACLY^WT and ACLY^3KR. Naive CD4+ T cells transfected with GFP-ACLY^WT or -ACLY^3KR were polarized as in *Figure 1A* for 24 hr. (**F**) Extracts from cells pre-treated with MG132 were assayed with immunoprecipitation (IP) for ACLY ubiquitination. IgG serves as a negative control. (**G**) WB analysis for ACLY protein. (**H**) Cell lysates were prepared for the analysis of carnitine palmitoyltransferase 1 (CPT1) activity. (**I**) ACLY ubiquitination impacts on iTreg differentiation. Cells transfected with GFP-ACLY^WT or -ACLY^3KR were polarized as in *Figure 1A* and assessed with flow cytometry (FCM) (left). Quantification of GFP+CD25+Foxp3+ iTreg cells (right). (**A–C, G–I**) Quantification shows mean ± SD based on three independent experiments. Student's t-test (**A**), two-way analysis of variance (ANOVA) test (**B**), or one-way ANOVA test (**C, G–I**) was used. *p<0.05,

*Figure 3 continued on next page*

Figure 3 continued

**p<0.01, ***p<0.001, and ****p<0.0001; ns, nonsignificant. For WB in (A–G), one representative experiment out of three is represented. Associated scores indicate mean intensities based on three biological replicas.

The online version of this article includes the following figure supplement(s) for figure 3:

**Figure supplement 1.** Assessment of ACLY activity and mRNA in Th0 and iTreg cells.
**Figure supplement 2.** K530/536/544 are important for ACLY degradation.

supplement 3A–B). Due to the lack of CUL3, the frequency of CD4$^+$Foxp3$^+$ Treg cells in colon was reduced as expected (*Figure 4—figure supplement 3C*). Notably, when naive CD4$^+$ T cells from these CUL3-deficient mice were induced into iTreg cells in vitro, both ACLY ubiquitination and degradation induced by TGFβ1 were blocked (*Figure 4F–H*). These observations fully recapitulated what we previously found in siRNA-mediated CUL3-depleted cells. More importantly, alterations induced by CUL3 deficiency, including increase in malonyl-CoA level and decrease in CPT1 activity and iTreg differentiation, were dampened by ACLY depletion (*Figure 4—figure supplement 3D*, *Figure 4I–J*). Together, TGFβ1 induces CUL3-KLHL25-mediated ACLY ubiquitination and degradation for iTreg differentiation.

It is worth mentioning that the suppressive function of iTreg cells did not seem to be affected by ACLY ubiquitination and degradation. When we knocked down *Cul3* and/or *Acly* with targeting siRNAs, levels of distinct surrogate molecules typically associated with the suppressive activity of iTreg cells, including CTLA4, ICOS, TIGIT, and IL-10, remained rather stable (*Figure 4—figure supplement 4A*). In agreement with this observation, we took the advantage of the available *Foxp3$^{YFP-Cre}$* mice (*Priyadharshini et al., 2018*) and induced YFP-iTreg cell differentiation under various conditions in vitro. We found that YFP-iTreg cells lacking CUL3 alone or in combination with ACLY could still retain the potent suppressive function (*Figure 4—figure supplement 4B*). Hence, instead of the suppressive function of iTreg, ACLY ubiquitination and degradation mainly affect iTreg differentiation.

## CUL3-KLHL25-ACLY ubiquitination axis regulates human iTreg differentiation

To further corroborate the role of CUL3-KLHL25-mediated ACLY ubiquitination in human iTreg differentiation, we isolated naive CD4$^+$ T cells from human peripheral blood and induced them into either Th0 or iTreg cells. As expected, the level of ACLY in iTreg dropped significantly as compared with Th0 (*Figure 5A*). ACLY overexpression and inhibition resulted in downregulation and upregulation in iTreg differentiation, respectively (*Figure 5B–C*). Furthermore, TGFβ1 treatment promoted CUL3-KLHL25 association with ACLY, resulting in ACLY ubiquitination and degradation (*Figure 5D–F*). Upon the mutation of K540/546/554 in ACLY (ACLY$^{3KR}$), TGFβ1-induced ACLY ubiquitination and degradation, as well as iTreg differentiation, were remarkably dampened (*Figure 5E–G*). Taken together, these data corroborate the important role of CUL3-KLHL25-mediated ACLY ubiquitination in human iTreg differentiation.

## ACLY inhibition alleviates mouse colitis by promoting iTreg cell generation

As a common digestive system disease, IBD poses a serious threat to public health, and the number of patients with IBD keeps growing (*Bernstein et al., 2010*). Manipulation of iTreg functions holds promise for the therapeutic treatment of IBD (*Clough et al., 2020*; *Izcue et al., 2006*; *van Herk and Te Velde, 2016*). Next, we performed a functional assay for CUL3-KLHL25-mediated ACLY ubiquitination in a classic mouse IBD model (*Wirtz and Neurath, 2007*). Same amount of naive CD4$^+$ T cells was isolated from wild-type (WT) or *Cd4$^{Cre}$Cul3$^{fl/fl}$* (*Cul3$^{-/-}$*) mice and transfected with siRNAs prior to polarization to obtain Th0 (WT), iTreg (WT), iTreg (*Cul3$^{-/-}$*), or iTreg (*Cul3$^{-/-}$*+si*Acly*) cells (*Figure 6A*). Compared to control, the lack of CUL3 impaired iTreg differentiation, while ACLY depletion compromised CUL3-deficiency-induced changes (*Figure 6—figure supplement 1*, *Figure 6B*). Upon the adoptive transfer of these cells along with WT-mice-derived naive CD4$^+$ T cells used to induce colitis into *Rag1$^{-/-}$* recipient mice (*Figure 6A*), Th0 (WT) group suffered serious colitis lying in body weight loss, disease activity index (DAI) score increase, colon length shortening,

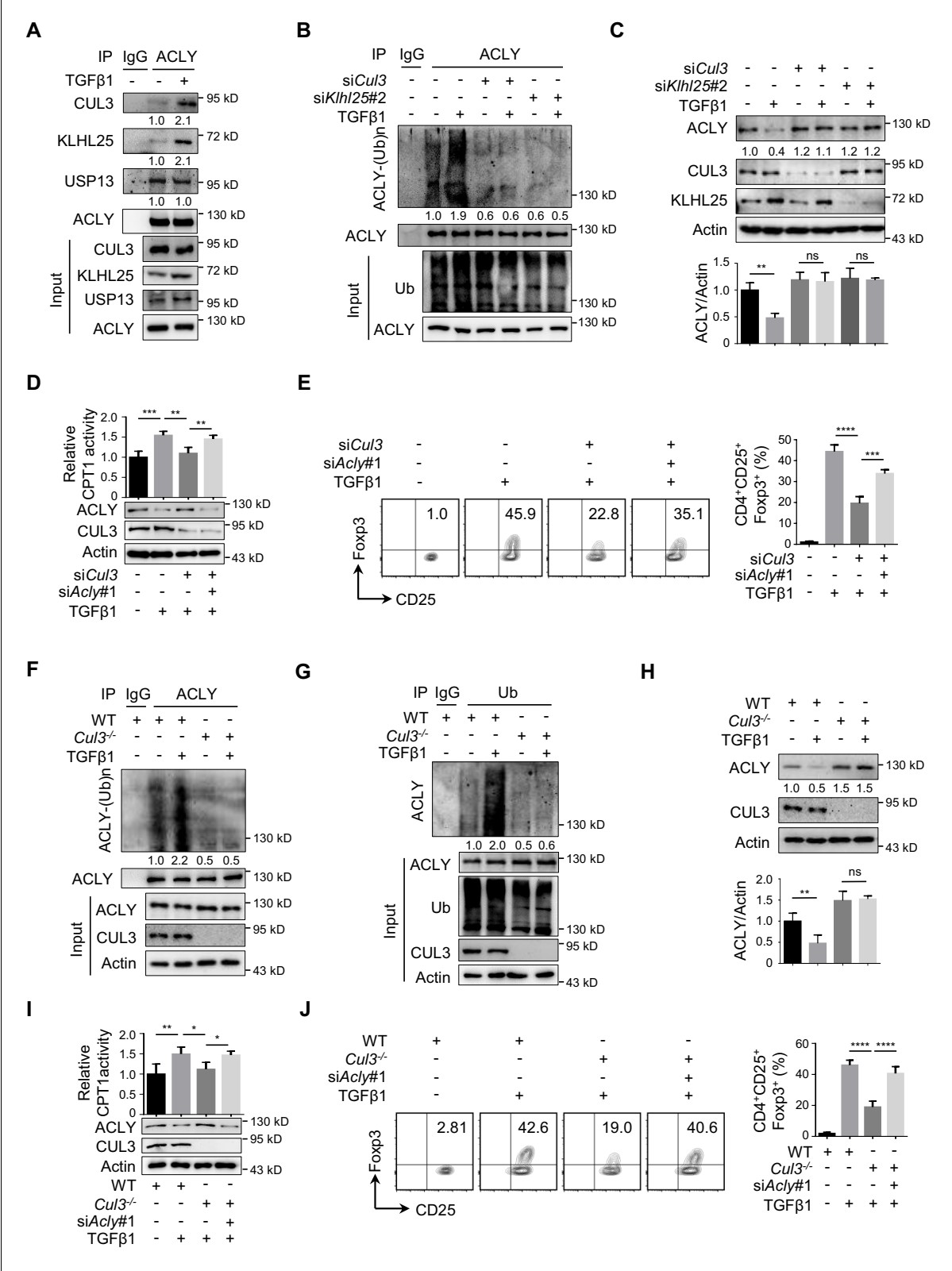

**Figure 4.** CUL3-KLHL25 is responsible for TGFβ1-mediated ACLY ubiquitination. (**A**) Assay of ATP-citrate lyase (ACLY) interaction with CUL3-KLHL25. Cells polarized as in *Figure 1A* for 24 hr were treated with MG132 and assayed by immunoprecipitation (IP) with ACLY antibody. IgG serves as a negative control. (**B–E**) Functional assessment of CUL3-KLHL25 depletion in inducible regulatory T (iTreg) differentiation. Naive CD4⁺ T cells transfected with desired small interfering RNAs (siRNAs) were polarized as in *Figure 1A* for 24 hr (**B–D**) or 72 hr (**E**). They were pre-treated with MG132 before IP for

*Figure 4 continued on next page*

*Figure 4 continued*

the detection of ACLY ubiquitination (**B**), analyzed by western blotting (WB) with indicated antibodies (**C**), evaluated for carnitine palmitoyltransferase 1 (CPT1) activity (**D**), and stained for CD4/CD25/Foxp3 to identify iTreg cells prior to flow cytometry (FCM) analysis (**E**). (**F–H**) Functional assays for ACLY ubiquitination in cells derived from *Cul3* knockout mice. Naive CD4$^+$ T cells isolated from wild-type (WT) or *Cd4$^{Cre}$Cul3$^{fl/fl}$* mice were polarized as in *Figure 1A* for 24 hr. Cells were pre-treated with MG132 before IP with antibodies against ACLY (**F**) or ubiquitin (Ub) (**G**), or analyzed for ACLY expression with WB (**H**). (**I, J**) Impact of CUL3 deficiency on fatty acid oxidation (FAO) and iTreg differentiation. Naive CD4$^+$ T cells transfected with siRNAs were induced as in *Figure 1A* for 24 hr (**I**) or 72 hr (**J**). Cells were lysed for CPT1 activity detection (**I**) or stained for CD4/CD25/Foxp3 to identify iTreg cells prior to FCM analysis (**J**). (**C–E, H–J**) Quantification shows mean ± SD based on three independent experiments, with significance determined by one-way analysis of variance (ANOVA) test. *p<0.05, **p<0.01, ***p<0.001, and ****p<0.0001; ns, nonsignificant. For WB in (**A–D, F–I**), one representative experiment out of three is represented. Associated scores indicate mean intensities based on three biological replicas.

The online version of this article includes the following figure supplement(s) for figure 4:

**Figure supplement 1.** Examination of ACLY association with ubiquitin ligases and deubiquitinase.

**Figure supplement 2.** Knockdown of CUL3-KLHL25 blocks TGFβ1-induced ACLY degradation.

**Figure supplement 3.** Analysis of Treg number in the intestines of *Cd4$^{Cre}$Cul3$^{fl/fl}$* mice.

**Figure supplement 4.** Functional analysis of ACLY ubiquitination for iTreg cells.

inflammatory cell infiltration, mucosal edema and injury as well as crypt damage increase (*Figure 6C–E*). As expected, iTreg (WT) mice showed significantly alleviated pathological alterations associated with colitis (*Figure 6C–E*). When we injected recipient mice with iTreg (*Cul3$^{-/-}$*) cells, less alleviation in colitis was observed as compared to that in iTreg (WT) group (*Figure 6C–E*). Importantly, knocking down *Acly* in iTreg (*Cul3$^{-/-}$*+si*Acly*) cells prior to the adoptive transfer rescued CUL3-deficiency-induced failures in colitis alleviation (*Figure 6C–E*). Using the same T-cell adoptive transfer strategy, we found that direct inhibition of ACLY by SB204990, which led to elevation in iTreg population from both mesenteric lymphatic node (MLN) and colonic lamina propria (cLP) and alleviation of dextran sodium sulfate (DSS)-induced colitis (*Figure 6—figure supplement 2A–F*), also effectively relieved T-cell-transfer-induced colitis in mice (*Figure 6—figure supplement 3A–E*). Together, these in vivo data confirm the important role of CUL3-KLHL25-mediated ACLY ubiquitination in colitis alleviation.

In addition, we also investigated the role of CUL3-dependent ACLY regulation in an alternative inflammatory disease associated with deregulated Treg function, namely allergic diarrhea (*Kordowski et al., 2019*; *Wang et al., 2018*; *Yamamoto et al., 2019*). Adoptive transfer of WT iTreg cells, but not CUL3-deficient cells (*Cul3$^{-/-}$*), into recipient mice remarkably alleviated the defects associated with ovalbumin (OVA)-induced allergic diarrhea, including upregulation of OVA-specific IgE antibodies and mastocyte protease (MCPT-1) in serum (*Kordowski et al., 2019*; *Figure 6—figure supplement 4A–E*). Knockdown of *Acly* rescued the effect of CUL3 deficiency on diarrhea (*Figure 6—figure supplement 4B–E*). Hence, these results suggest an important role for 'CUL3-ACLY' axis in the regulation of diarrhea.

## Discussion

iTreg cells are essential for the regulation of immune homeostasis, and their establishment relies on TGFβ1 induction (*Chen et al., 2003*; *Dominguez-Villar and Hafler, 2018*). Recent findings suggest that a shift from FAS to FAO is required for efficient iTreg differentiation (*Berod et al., 2014*; *Michalek et al., 2011*), but the control of this process is still unclear. Here we unveiled a TGFβ1-CUL3-KLHL25-ACLY axis that is important for the metabolic reprogramming in fatty acid metabolism during iTreg differentiation.

As one of the most well-studied signaling pathways, TGFβ1 governs a number of signaling cascades, including canonical Smad signals and non-canonical MAPK (ERK/JNK/P38), PI3K/Akt, and Rho-like GTPase signals, mainly by regulating different kinases to phosphorylate their downstream factors (*Vander Ark et al., 2018*; *Zhang, 2017*). In the process of iTreg differentiation, TGFβ1 activates TGFRI/II, which are transmembrane protein serine/threonine kinases and directly phosphorylate Smad2/3, to improve Foxp3 expression (*Schlenner et al., 2012*). Interestingly, our work here demonstrated that, through ubiquitination of a key metabolic enzyme, TGFβ1 launched a reprogramming in fatty acid metabolism and promoted iTreg differentiation. These findings indicate

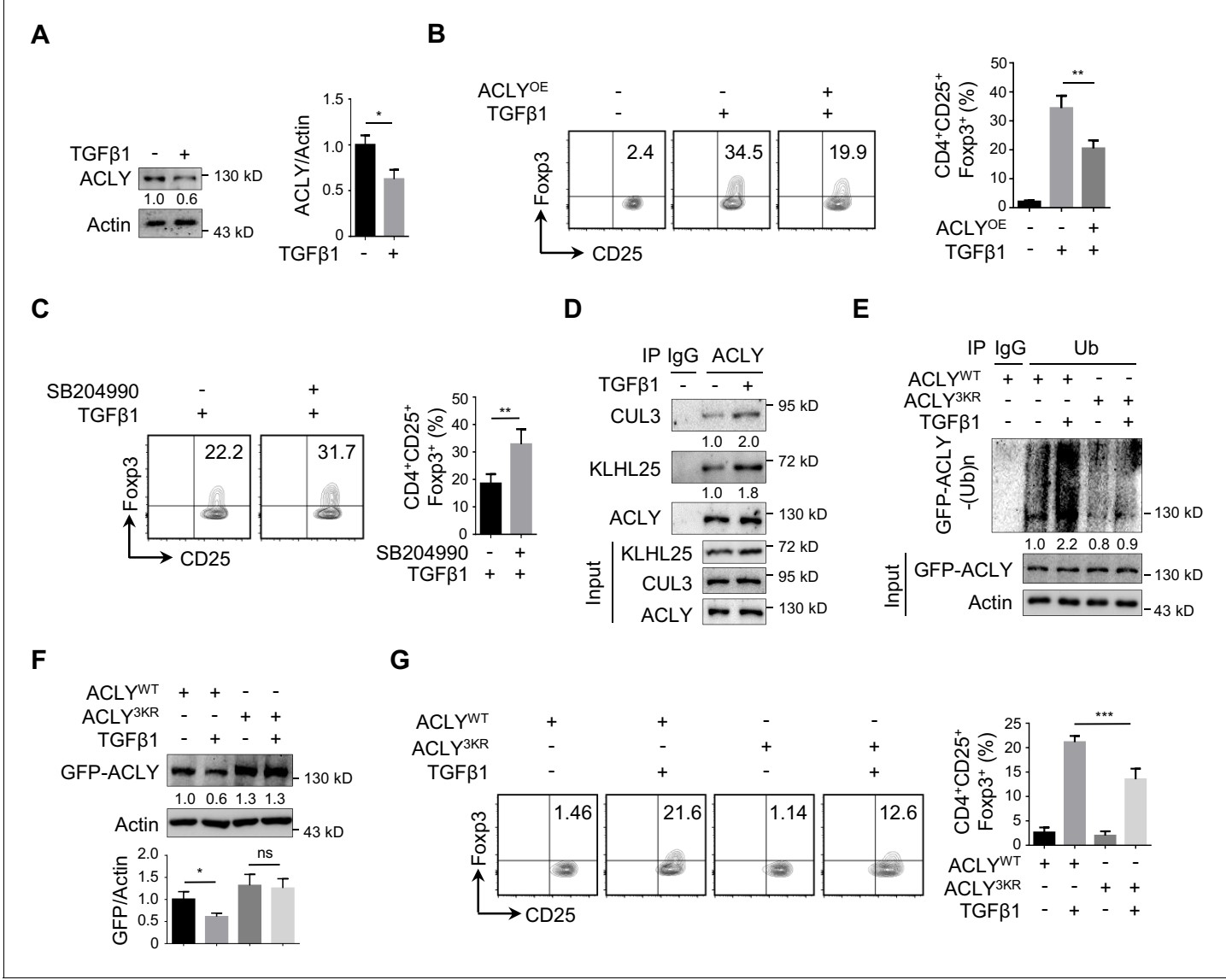

**Figure 5.** CUL3-KLHL25-ACLY ubiquitination axis regulates human iTreg differentiation. (A) Assay of ATP-citrate lyase (ACLY) expression in human inducible regulatory T (iTreg) cells. Naïve CD4+ T cells isolated from human peripheral blood were cultured with Dynabeads Human T-Activator CD3/CD28 and rhIL-2 to induce activated T (Th0) cells or simultaneously supplemented with rhTGFβ1 (2 ng/ml) to induce iTreg cells for 24 hr. Cell extracts were subsequently analyzed for ACLY expression by western blotting (WB). (B) ACLY overexpression affects iTreg cell differentiation. Human naïve CD4+ T cells transfected with GFP-ACLYWT were polarized as in (A) for 72 hr prior to the assessment of iTreg cells by flow cytometry (FCM) (left). Quantification of iTreg populations (right). (C) ACLY inhibition affects iTreg differentiation. Cells treated with SB204990 were cultured under Th0- or iTreg (with 0.5 ng/ml rhTGFβ1)-polarization condition for 72 hr and subjected to the analysis (left) and quantification (right) of iTreg cells. (D) ACLY interaction with CUL3-KLHL25. Cells prepared as in (A) were pre-treated with MG132 before immunoprecipitation (IP) with ACLY antibody. IgG serves as a negative control. (E–G) ACLY ubiquitination regulates human iTreg differentiation. Cells transfected with GFP-ACLYWT or -ACLY3KR were induced under Th0- or iTreg-polarization condition as described in (A) for 24 hr (E, F) or 72 hr (G). Cells were pre-treated with MG132 before IP for the assessment of protein ubiquitination (E), examined by WB (F), or evaluated for GFP/CD25/Foxp3 expression with FCM (left), and quantified (right) (G). (A–C, F, G) Quantification shows mean ± SD based on three independent experiments, with significance determined by Student's t-test (A, C) or one-way analysis of variance (ANOVA) test (B, F, G). *p<0.05, **p<0.01, ***p<0.001; ns, nonsignificant. For WB in (A) and (D–F), one representative experiment out of three is represented. Associated scores indicate mean intensities based on three biological replicas.

ubiquitination as a useful means for TGFβ1 to coordinate fatty acid metabolism and induce iTreg differentiation.

Moreover, CUL3-KLHL25 is a part of the biggest ubiquitin ligase Cullin-Ring family and is involved in the regulation of numerous biological processes, such as cell cycle regulation, DNA damage

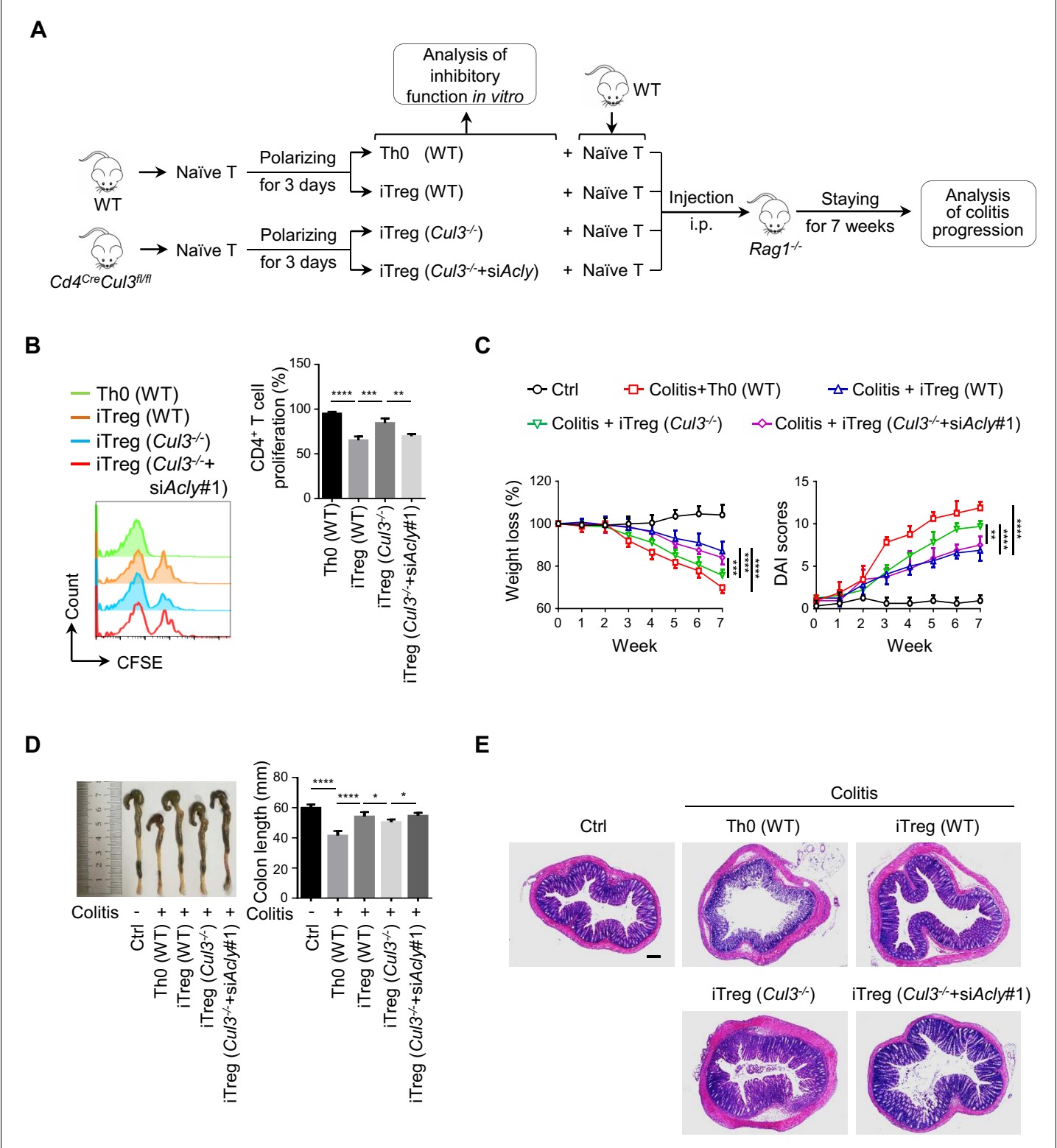

**Figure 6.** ACLY ubiquitination promotes iTreg cell generation and alleviates mouse colitis. (A) Experimental design. (B) Impact of inducible regulatory T (iTreg) cells on T-cell proliferation. Activated T (Th0) (wild-type, WT), iTreg (WT), iTreg (*Cul3$^{-/-}$*), and iTreg (*Cul3$^{-/-}$*+si*Acly*#1) cells were cocultured with carboxyfluorescein diacetate succinimidyl ester (CFSE)-stained naive CD4$^+$ T cells and treated with Dynabeads Mouse T-Activator CD3/CD28 for 72 hr. T-cell proliferation was evaluated using flow cytometry (FCM) (left). Quantification analysis for CD4$^+$ T-cell proliferation (right) based on three independent experiments. One-way analysis of variance (ANOVA) test, mean ± SD. **p<0.01, ***p<0.001, ****p<0.0001. (C–E) Systematic evaluation of

*Figure 6 continued on next page*

*Figure 6 continued*

colitis in mice adoptively transferred with different cells. (**C**) Body weight (left) and disease activity index (DAI) score (right) based on body weight loss, stool consistency, and blood in the stool were recorded weekly. 7 weeks later, entire colons from mice treated in different ways were removed for length assessment (**D**) and hematoxylin and eosin (H and E) staining, which is used for revealing histopathological changes (**E**). Scale bar, 200 μm. One representative experiment out of four biological replicas is represented. (**C, D**) Data represent mean ± SD (n = 4), with significance determined by two-way ANOVA test (**C**) or one-way ANOVA test (**D**). *p<0.05, **p<0.01, ***p<0.001, ****p<0.0001.

The online version of this article includes the following figure supplement(s) for figure 6:

**Figure supplement 1.** ACLY knockdown influences the differentiation of naive CD4[+] T cells derived from *Cul3* knockout mice.

**Figure supplement 2.** SB204990 administration elevates iTreg population in vivo and alleviates mice colitis.

**Figure supplement 3.** ACLY inhibition promotes iTreg differentiation and relieves mice colitis.

**Figure supplement 4.** ACLY ubiquitination is involved in the regulation of iTreg differentiation and allergic diarrhea in mice.

responses, and cell apoptosis (*Davidge et al., 2019*; *Genschik et al., 2013*). The lack of CUL3 has been associated with defects in embryogenesis, hypertension, diminished angiogenesis, progressive interstitial inflammation, exacerbated colonic inflammation, and swelled spleens and lymph nodes (*Agbor et al., 2019*; *Li et al., 2017*; *Maekawa et al., 2017*; *Mathew et al., 2012*; *Saritas et al., 2019*; *Singer et al., 1999*). Interestingly, some of these defects, for example, progressive interstitial inflammation, exacerbated colonic inflammation, and swelled spleens and lymph nodes, are clearly relevant to immune system imbalance. In this study, the functional characterization of CUL3-KLHL25 in iTreg differentiation implies that disturbed iTreg cell homeostasis may contribute to CUL3-deficiency-associated defects, in particular, inflammatory-related phenomena. In addition, this may provide a new angle for re-evaluating and understanding severe pathological changes induced by the loss of CUL3.

It is also worth mentioning that the loss of CUL3 did not affect the differentiation of Th1, Th2, or Th17 in vitro, but enhanced follicular helper T (Tfh) cell number and responses in the germinal center (GC). These observations and our findings suggest that CUL3 may play distinct roles in different types of T cells, such as Tfh and Treg cells. In GC, Tfh cells can be suppressed by follicular regulatory T (Tfr) cells, which are also crucial for the maintenance of immune homeostasis (*Essig et al., 2017*; *Huang et al., 2019*). This specific type of T cells is actually derived from Treg cells (*Essig et al., 2017*; *Huang et al., 2019*). Whether CUL3 could finetune the balance between Tfh and Tfr for the control of GC function remains to be addressed in future.

Growing evidence shows that the regulation of cell metabolism is vital for T-cell proliferation, differentiation, and functions (*Galgani et al., 2015*; *Guo, 2017*; *Maciolek et al., 2014*; *Wang and Solt, 2016*). Different types of T cells are usually associated with distinct metabolic status and features. Unlike effector T cells, including Th1, Th2, and Th17, which 'prefer' FAS, iTreg cells 'favor' FAO (*Maciolek et al., 2014*; *Michalek et al., 2011*; *Wang et al., 2011*). During iTreg differentiation, suppressing FAS could probably be advantageous to keep the differentiation process on the right track and prevent differentiating cells from rushing into unwanted pathways. Also, we found that downregulation of FAS reduced malonyl-CoA, the potent physiological inhibitor of CPT1, and this could facilitate the unleashing of CPT1 activity for FAO and thereby iTreg differentiation. Therefore, reprogramming in fatty acid metabolism, in particular, the switch from FAS to FAO, is vital for the efficient differentiation of iTreg cells.

Although the essential role of FAO and CPT1 in iTreg differentiation has been documented (*MacIver et al., 2013*; *Michalek et al., 2011*; *Patel and Powell, 2017*; *Zhang et al., 2017*), a more recent investigation challenged this conclusion and proposed that CPT1 is dispensable for iTreg differentiation (*Raud et al., 2018*). *Raud et al., 2018* found that naive CD4[+] T cells isolated from mice with CPT1 deletion in T cells (TCPT1 mice) did not seem to have any defect in iTreg differentiation and proposed that the phenotypes induced by high-dose CPT1 inhibitor, etomoxir (>100 μM), were due to the off-target effect. However, we found that siRNA-mediated *Cpt1* knockdown or low-dose etomoxir (5 μM), which inhibits CPT1 without introducing off-target effects, impaired iTreg differentiation in vitro (*Hao et al., 2021*; *O'Connor et al., 2018*; *Figure 2F*, *Figure 2—figure supplement 5B*). Of note, TCPT1 mice reported in *Raud et al., 2018* survived from the lack of CPT1. One thus can envision that various metabolic genes/regulators have to be re-adjusted to compensate for the loss of CPT1, and metabolic homeostasis needs to be re-established. As a consequence, the defect (s) associated with the loss of CPT1 would be gradually compensated or masked during mouse

development. In contrast, blocking CPT1 function by a specific inhibitor or siRNA that mimics an acute disruption in FAO might be useful for minimizing long-term metabolic adaption and directly evaluating CPT1 function in iTreg differentiation. In addition, our in vitro functional assays showed that alterations in CPT1 activity induced by ACLY and/or CUL3 depletion failed to change the suppressive function of iTreg cells (*Figure 4D,I*, *Figure 4—figure supplement 4A–B*). In a recent study, *Saravia et al., 2020* also showed that the deletion of CPT1 in Foxp3[+] Treg cells did not influence cell proliferation or function (*Saravia et al., 2020*). Together, these findings argue that CPT1 may be dispensable for the regulation and maintenance of iTreg function.

FAS is controlled by a series of enzymes, three of which are rate-limiting enzymes, namely ACLY, ACC, and FASN. During iTreg differentiation, only ACLY changed significantly. These observations suggest that, rather than ACC and FASN, ACLY is apt to function as a sensor for FAS to respond to TGFβ1 signal. ACLY is responsible for the conversion of citrate into acetyl-CoA and OAA. Its enzymatic activity and expression are tightly coupled with the level of citrate, which could block glycolysis by inhibiting phosphofructokinase (PFK) (*Iacobazzi and Infantino, 2014*; *Zaidi et al., 2012*). Notably, inhibition of glycolysis has a positive impact on iTreg differentiation (*Shi et al., 2011*). Therefore, TGFβ1-induced downregulation of ACLY, which is probably accompanied by citrate accumulation, might also contribute to the repression of glycolysis for iTreg differentiation.

Because of the great potential as a target for pharmaceutical intervention against hyperlipidemia, hypercholesterolemia, obesity, and cancer, ACLY has attracted considerable amount of attention over the recent decade (*Feng et al., 2020*; *Migita et al., 2008*). A growing list of ACLY inhibitors, including bempedoic acid (BemA), SB204990, curcumin, cinnamon polyphenol, and hydroxycitric acid, is under extensive investigation. It is worth mentioning that BemA, a small molecule inhibiting liver-specific ACLY, has been tested in a phase-three clinical trial for the control and treatment of hypercholesterolemia (*Feng et al., 2020*; *Pinkosky et al., 2016*). Results in this work have shown that ACLY inhibition by SB204990 effectively alleviated colitis in mice, suggesting that this small-molecule inhibitor (SB204990) may have a promising role in the clinical treatment for IBD. Thus, future investigations systematically evaluating ACLY as a drugable target for IBD and other immune diseases are in need.

Functional characterization of CUL3-KLHL25-mediated ACLY ubiquitination pinpoints ACLY as a regulatory node integrating TGFβ1 signaling with iTreg differentiation and therefore brings insights into the modulation and maintenance of immune homeostasis. Moreover, ACLY inhibitors with pharmaceutical potentials are apparently of interest for immunotherapy of autoimmunity and unwanted inflammation and certainly require further investigations.

# Materials and methods

**Key resources table**

| Reagent type (species) or resource | Designation | Source or reference | Identifiers | Additional information |
|---|---|---|---|---|
| Strain, strain background (*Mus musculus*) | C57BL/6J | Beijing HFK Bioscience | | |
| Strain, strain background (*M. musculus*) | Cd4[Cre] | *Dong et al., 2008* | | |
| Strain, strain background (*M. musculus*) | Cul3[fl/fl] | The Jackson Laboratory | Stock #: 028349 | |
| Strain, strain background (*M. musculus*) | Cd4[Cre]Cul3[fl/fl] | This paper | | |
| Strain, strain background (*M. musculus*) | Foxp3[YFP-Cre] | The Jackson Laboratory | Stock #: 016959 | |

*Continued on next page*

*Continued*

| Reagent type (species) or resource | Designation | Source or reference | Identifiers | Additional information |
|---|---|---|---|---|
| Strain, strain background (*M. musculus*) | B6/*Rag1*$^{-/-}$ | Jiangsu Gempharmatech | Stock #: T011699 | |
| Cell line (*Homo sapiens*) | HEK293T | ATCC | Cat. #: ACS-4500 | |
| Biological sample (*Homo sapiens*) | Healthy adult peripheral blood | Jilin blood center | | |
| Antibody | Mouse anti-CD3/CD28 beads | Gibco | Cat. #: 11453D | T-cell activation (beads:cells, 1:1) |
| Antibody | Human anti-CD3/CD28 beads | Gibco | Cat. #: 11132D | T-cell activation (beads:cells, 1:1) |
| Antibody | Anti-mouse CD4-FITC | BD Biosciences | Cat. #: 553046 RRID:AB_394582 | FCM (1:200) |
| Antibody | Anti-mouse CD25-PE | BD Biosciences | Cat. #: 561065 RRID:AB_10563211 | FCM (1:200) |
| Antibody | Anti-mouse Foxp3-Alexa Fluor 647 | BD Biosciences | Cat. #: 560401 RRID:AB_1645201 | FCM (1:200) |
| Antibody | Anti-human CD4-FITC | Biolegend | Cat. #: 357406 RRID:AB_2562357 | FCM (1:200) |
| Antibody | Anti-human CD25-PE | Biolegend | Cat. #: 302605 RRID:AB_314275 | FCM (1:200) |
| Antibody | Anti-human Foxp3-Alexa Fluor 647 | Biolegend | Cat. #: 320114 RRID:AB_439754 | FCM (1:200) |
| Antibody | Anti-mouse CTLA4-PE-Cy7 | Biolegend | Cat. #: 106311 RRID:AB_10901170 | FCM (1:200) |
| Antibody | Anti-mouse ICOS-PE-Cy7 | Biolegend | Cat. #: 117421 RRID:AB_2860636 | FCM (1:200) |
| Antibody | Anti-mouse TIGIT-PE-Cy7 | Biolegend | Cat. #: 142107 RRID:AB_2565648 | FCM (1:200) |
| Antibody | Anti-mouse IL-10-PE-Cy7 | Biolegend | Cat. #: 505025 RRID:AB_11149682 | FCM (1:200) |
| Antibody | Anti-ACLY | Abcam | Cat. #: Ab40793 RRID:AB_722533 | IP (1:100) WB (1:1000) |
| Antibody | Anti-eGFP | Proteintech | Cat. #: 66002–1-Ig RRID:AB_11182611 | IP (1:100) WB (1:1000) |
| Antibody | Anti-Actin | Proteintech | Cat. #: 66009–1-Ig RRID:AB_2687938 | WB (1:1000) |
| Antibody | Anti-CUL3 | Santa Cruz | Cat. #: sc-166110 RRID:AB_2245478 | IP (1:50) WB (1:200) |
| Antibody | Anti-KLHL25 | Santa Cruz | Cat. #: sc-100774 RRID:AB_1124139 | IP (1:50) WB (1:200) |
| Antibody | Anti-UBR4 | Abcam | Cat. #: Ab86738 RRID:AB_1952666 | IP (1:100) WB (1:1000) |
| Antibody | Anti-USP13 | Santa Cruz | Cat. #: sc-514416 | IP (1:50) WB (1:200) |
| Antibody | Anti-Ubiquitin | Abcam | Cat. #: ab7780 RRID:AB_306069 | IP (1:100) WB (1:1000) |
| Antibody | Anti-Lamin b1 | CST | Cat. #: 12586 | WB (1:1000) |
| Antibody | Anti-β-Tubulin | Sigma | Cat. #: T0198 RRID:AB_2210695 | WB (1:1000) |
| Antibody | Anti-ATP5A1 | Bimake | Cat. #: A5288 | WB (1:1000) |

*Continued*

| Reagent type (species) or resource | Designation | Source or reference | Identifiers | Additional information |
|---|---|---|---|---|
| Antibody | Anti-Pan-acetylation | CST | Cat. #: 9441 RRID:AB_331805 | WB (1:1000) |
| Antibody | Goat anti-Rabbit | Signalway Antibody | Cat. #: L3012-2 | WB (1:10,000) |
| Antibody | Goat anti-Mouse | Signalway Antibody | Cat. #: L3032-2 | WB (1:10,000) |
| Commercial assay or kit | Mouse CD4$^+$ Naïve T Cell isolation kit | Biolegend | Cat. #: 480040 | |
| Commercial assay or kit | Human CD4$^+$ Naïve T Cell isolation kit | Biolegend | Cat. #: 480042 | |
| Commercial assay or kit | Mouse T cell nucleofection kit | Lonza | Cat. #: VPA1006 | |
| Commercial assay or kit | Mouse mevalonate ELISA kit | Sino Best Biological Technology | Cat. #: YX-132201M | |
| Commercial assay or kit | Mouse MCPT1 ELISA kit | Sino Best Biological Technology | Cat. #: YX-130316M | |
| Commercial assay or kit | Mouse OVA-specific IgE ELISA kit | Sino Best Biological Technology | Cat. #: YX-152201M | |
| Commercial assay or kit | Mouse mevalonate-5-pyrophosphate ELISA kit | Shanghai Ruifan Biological Technology | Cat. #: RF8872 | |
| Commercial assay or kit | Mouse HMG-CoA ELISA kit | Shanghai Ruifan Biological Technology | Cat. #: RF8726 | |
| Commercial assay or kit | Mitochondrial isolation kit | Beyotime Biotechnology | Cat. #: C3601 | |
| Commercial assay or kit | Cell Nuclear/ Cytoplasm Protein Isolation Kit | Beyotime Biotechnology | Cat. #: P0028 | |
| Commercial assay or kit | ACLY activity assay kit | Suzhou comin Biotechnology | Cat. #: ACL-1-Y | |
| Commercial assay or kit | ACC activity assay kit | Suzhou comin Biotechnology | Cat. #: ACC-1-Y | |
| Commercial assay or kit | FASN activity assay kit | Suzhou comin Biotechnology | Cat. #: FASN-1-Y | |
| Commercial assay or kit | CPT1 activity assay kit | Suzhou comin Biotechnology | Cat. #: CPT1-1-Y | |
| Commercial assay or kit | Acetyl-CoA Fluorometric Assay kit | BioVision | Cat. #: K317-100 | |
| Commercial assay or kit | OAA assay kit | Sigma | Cat. #: MAK070 | |
| Commercial assay or kit | Transcriptional factor buffer set | BD | Cat. #: 562574 | |
| Commercial assay or kit | CFSE | Biolegend | Cat. #: 423801 | 1 µM |
| Commercial assay or kit | CellTrace Violet | Thermo Fisher | Cat. #: C34557 | 1 µM |
| Commercial assay or kit | Mito Stress Test kits | Agilent | Cat. #: 103010–100 | |
| Commercial assay or kit | Seahorse XF Palmitate-BSA FAO substrate | Agilent | Cat. #: 102720–100 | |

*Continued on next page*

*Continued*

| Reagent type (species) or resource | Designation | Source or reference | Identifiers | Additional information |
|---|---|---|---|---|
| Recombinant DNA reagent | pWPXLD-mACLY (plasmid) | This paper | | |
| Recombinant DNA reagent | pWPXLD-mACLY$^{3KR(K530/536/544)}$ (plasmid) | This paper | | |
| Recombinant DNA reagent | pWPXLD-hACLY (plasmid) | This paper | | |
| Recombinant DNA reagent | pWPXLD-hACLY$^{3KR(K540/546/554)}$ (plasmid) | This paper | | |
| Peptide, recombinant protein | Recombinant mouse IL-2 | Peprotech | Cat. #: 212–12 | 7.5 ng/ml |
| Peptide, recombinant protein | Recombinant human IL-2 | Gibco | Cat. #: PHC0026 | 7.5 ng/ml |
| Peptide, recombinant protein | Recombinant human TGFβ1 | Peprotech | Cat. #: 100-21-10 | 0.5 or 2 ng/ml |
| Chemical compound, drug | SB204990 | Tocris Bioscience | Cat. #: 154566-12-8 | 100 μM |
| Chemical compound, drug | MG132 | Selleck | Cat. #: S2619 | 10, 20, or 40 μM |
| Chemical compound, drug | CHX | Tocris Bioscience | Cat. #: 0970 | 10 μg/ml |
| Chemical compound, drug | Leupeptin | Selleck | Cat. #: S7380 | 0.4 or 0.8 mM |
| Chemical compound, drug | Dextran sulfate sodium (36,000–50,000) | MP Biomedicals | Cat. #: 0216011080 | 2% |
| Chemical compound, drug | PF05175157 | TOCRIS | Cat. #: 5709 | 5 μM |
| Chemical compound, drug | Cerulenin | Biovision | Cat. #: 1579–5 | 4 μM |
| Software, algorithm | FlowJo 10.1 | FlowJo LLC | https://www.flowjo.com/ | |
| Software, algorithm | GraphPad Prism 6 | GraphPad Software | https://www.graphpad.com | |
| Other | RPMI 1640 | Sigma-Aldrich | Cat. #: R66504 | |
| Other | Fetal bovine serum | Biological Industries | Cat. #: 04-001-01 | |
| Other | XF Base Medium | Agilent | Cat. #: 103193–100 | |
| Other | OVA | Sigma-Aldrich | Cat. #: A5503 | |
| Other | Polybrene | Solarbio | Cat. #: H8761 | 8 μg/ml |

## Animals

C57BL/6J (B6) and B6/*Rag1*$^{-/-}$ mice were purchased from Beijing HFK Bioscience Co, Ltd, China, and Jiangsu Gempharmatech Co, Ltd, China, respectively. *Cul3*$^{fl/fl}$ mice were generously sponsored by

Jeffrey D Singer (Department of Biology, Portland State University, Portland, OR, USA.). $Cd4^{Cre}$ transgenic mice were generously provided by Zichun Hua (The State Key Laboratory of Pharmaceutical Biotechnology, Department of Biochemistry, College of Life Sciences, Nanjing University, Nanjing). $Cd4^{Cre}$ and $Cul3^{fl/fl}$ mice were crossed to generate mice specifically lacking CUL3 in their CD4$^+$ T cells. Polymerase chain reaction (PCR) primers used for amplifying $Cd4^{Cre}$ transgene and $Cul3^{fl/fl}$ mutant are shown in *Supplementary file 1*. $Foxp3^{YFP-Cre}$ mice were purchased from the Jackson laboratory.

Mice were bred and cohoused, four to six mice per cage, in a specific pathogen-free facility with a standard 12-hr alternate light/dark cycle at an ambient temperature of 22 ± 2℃ and 30–70% humidity at the Animal Research Center of Northeast Normal University (Changchun, China). Health status of mice was determined via daily observation by technicians supported by veterinary care. All mouse experiments were conducted in accordance with the protocols for animal use, treatment, and euthanasia approved by the local ethics committee (reference number: AP2019085, the institutional animal care and use committee (IACUC) of the Northeast Normal University).

## Induction of mouse iTreg in vitro

Naive CD4$^+$ T cells were isolated from the spleen of female mice at the age of 6–8 weeks with MojoSort Mouse CD4$^+$ Naïve T Cell Isolation Kit (Biolegend) according to the manufacturer's instructions. For Th0 induction, 2 to 3 × 10$^5$ naive CD4$^+$ T cells were cultured in Roswell Park Memorial Institute (RPMI) 1640 medium (Sigma-Aldrich) supplemented with 10% fetal bovine serum (FBS) (Biological Industries), penicillin-streptomycin (500 U; PAN biotech), and β-mercaptoethanol (50 µM; Sigma-Aldrich) and stimulated with Dynabeads Mouse T-Activator CD3/CD28 (Gibco) at a cell:bead ratio of 1:1 in the presence of rmIL-2 (200 U/ml; Peprotech) (*Wang et al., 2019*). To obtain iTreg cells, rhTGFβ1 (2 or 0.5 ng/ml; Peprotech) was simultaneously added along with the reagents used for inducing Th0 cells. ACLY inhibitor: SB204990 (Tocris Bioscience), ACC inhibitor: PF05175157 (Tocris Bioscience), FASN inhibitor: cerulenin (Biovision), protein synthesis inhibitor: CHX (Tocris Bioscience), proteasome inhibitor: MG132 (Selleck), lysosome inhibitor: leupeptin (Selleck).

## Induction of human iTreg in vitro

Human peripheral blood samples were obtained from Jilin Blood Center (Changchun, China). All work with human blood samples was approved by the local ethics committee (reference number: AP2019085, the ethical committee of the Northeast Normal University), and informed consent was obtained from all subjects. After Ficoll (GENVIEW) gradient, naive CD4$^+$ T cells were isolated with MojoSort human CD4$^+$ Naïve T Cell Isolation Kit (Biolegend) according to the manufacturer's instructions (*Ghosh et al., 2016*). For Th0 induction, 2 to 3 × 10$^5$ naive CD4$^+$ T cells were cultured in RPMI 1640 medium (Sigma) supplemented with 10% FBS (Biological Industries), penicillin-streptomycin (500 U; PAN biotech), and β-mercaptoethanol (50 µM; Sigma-Aldrich) and stimulated with Dynabeads human T-Activator CD3/CD28 (Gibco) at a cell:bead ratio of 1:1 in the presence of rhIL-2 (200 U/ml; Gibco). To obtain iTreg cells, rhTGFβ1 (2 or 0.5 ng/ml; Peprotech) was simultaneously added along with the reagents used for inducing Th0 cells.

## Flow cytometry analysis

Cells were collected in phosphate-buffered saline (PBS) containing 1% FBS (v/v) and fixed/permeabilized using the Transcription Factor Buffer Set (BD Biosciences) according to the manufacturer's instructions. For mouse iTreg analysis, selected protein markers were stained with fluorescein isothiocyanate (FITC) rat anti-mouse CD4, phycoerythrin (PE) rat anti-mouse CD25, and Alexa Fluor 647 rat anti-mouse Foxp3 monoclonal antibodies (BD Biosciences). For human iTreg analysis, protein markers were stained with FITC rat anti-human CD4, PE mouse anti-human CD25, and Alexa Fluor 647 mouse anti-human Foxp3 monoclonal antibodies (Biolegend).

## Quantitative real-time PCR

Total RNA was isolated using the TRIzol reagent (Invitrogen). Complementary DNA (cDNA) was synthesized with the PrimeScript RT reagent Kit plus gDNA Eraser (Takara) according to the manufacturer's instructions. Real-time PCR was performed on the QuantStudio 3 Real-Time PCR Instrument (Applied Biosystems) with a TB Green Premix Ex Taq (Tli RNaseH Plus) reagent (Takara).

The mRNA expression of genes was normalized to the expression of β-actin gene. Data were analyzed using the comparative cycling threshold method. Quantitative PCR primers are shown in *Supplementary file 1*.

### siRNA transfection

All siRNA transfections were performed with Mouse T-cell nucleofection kit (Lonza) according to the instructions using X-001 progress as described previously (*Mantei et al., 2008*). Cells were transiently transfected with siRNAs (siRNAs against *Acly*, *Cul3,* and *Klhl25* were commercially synthesized; Shanghai GenePharma) at a final concentration of 20 nM and rested in RPMI-1640 (with 10% FBS) for 4–5 hr before incubation under different conditions (Th0 or iTreg) for a desired period of time. siRNA sequences are shown in *Supplementary file 1*.

### Plasmid construction

DNA sequences coding mouse *Acly* gene (NCBI Accession No. NM_001199296.1) from mouse CD4$^+$ T cells and human *Acly* gene (NCBI Accession No. NM_001096.3) from human CD4$^+$ T cells were amplified by PCR and subsequently cloned into pWPXLd vector (containing green fluorescent protein (GFP) sequence) to construct pWPXLd-mACLY and pWPXLd-hACLY, respectively. K530, K536, and K544 mutations in mACLY and K540, K546, and K554 in hACLY were introduced by whole-plasmid PCR followed by DpnI digestion to obtain pWPXLd-mACLY$^{3KR\ (K530/K536/K544)}$ and pWPXLd-hACLY$^{3KR\ (K540/K546/K554)}$.

### Lentivirus transduction of CD4$^+$ T cells

HEK293T cells (without mycoplasma contamination) were grown in 10-cm cell-culture dishes to 70% confluence and were transfected with a three-plasmid system (pWPXLd-ACLY, PAX, PMD). The supernatant was harvested at 48, 72, and 96 hr after transfection and was concentrated 100-fold in an ultracentrifuge. Aliquots were stored at −80°C. Naive CD4$^+$ T cells were cultured under Th0-inducing condition as described above for 16–24 hr and were spin infected for two rounds with lentivirus supernatants (ACLY$^{WT}$, ACLY$^{3KR}$, or Empty) in the presence of 8 µg/ml polybrene (Solarbio) at 1000 $\times g$, 90 min, at 32°C as described previously (*Lin et al., 2019*; *Sun et al., 2015*). After 24 hr of infection, supernatants were removed and cells were selectively cultured under either Th0- or iTreg-inducing condition for a desired period of time.

### Assessment of metabolic molecules

For acetyl-CoA and OAA detection, cytosolic fractions from Th0 or iTreg cells were separated using Cell Mitochondria Isolation Kit (Beyotime). In brief, cells were washed with cold PBS, followed by homogenation for 20 s, and then centrifuged twice (first round, 600 $\times g$ for 5 min; second round, 11,000 $\times g$ for 5 min). Supernatants that represent cytosolic fractions without nuclei and mitochondria were confirmed by western blotting (WB) and deproteinized with a 10-kD spin filter by centrifuge for effluent preparation. Nuclear fraction was separated using Cell Nuclear/Cytoplasm Protein Isolation Kit (Beyotime) and also confirmed by WB. Cytosolic/nuclear acetyl-CoA and cytosolic OAA were assessed using PicoProbe acetyl-CoA fluorometric assay kit (Biovision) and OAA assay kit (Sigma), respectively, according to the manufacturer's protocol. For the analysis of malonyl-CoA, cells were cultured under iTreg-polarization condition for 24 hr and lysed with ice-cold 5% sulfosalicylic acid (Sigma-Aldrich) containing 50 µM dithioerythritol (DTE) (Sigma-Aldrich). After 600 $\times g$ centrifugation for 10 min at 4°C, the supernatants were collected for the detection of malonyl-CoA by high-performance liquid chromatography (HPLC). For the detection of HMG-CoA, mevalonate, and mevalonate-5-pyrophosphate, cells were homogenized in PBS and the supernatants were collected for detection of HMG-CoA, mevalonate, and mevalonate-5-pyrophosphate using HMG-CoA enzyme-linked immunosorbent assay (ELISA) kit (Shanghai Ruifan Biological Technology), Mevalonate ELISA kit (Sino Best Biological Technology), and Mevalonate-5-pyrophosphate ELISA kit (Shanghai Ruifan Biological Technology) according to the manufacturers' instructions. Equal amount of cells was used for different groups in the same assay.

## Extracellular flux analysis

For the FAO-associated oxygen consumption rate (OCR) assay by Seahorse XFp Analyzer (Agilent), Palmitate-BSA reagent (Agilent) and Mito Stress Test Kits (Agilent) were used according to the manufacturers' instructions with minor modifications. In brief, cells were collected and adhered to poly-D-lysine-coated XFp plates (4 to 6 $\times$ 10$^5$ cells/well) via centrifugation in serum-free XF Base Medium (Agilent). For evaluation of palmitate oxidation, cells were cultured with XF Base Medium (Agilent) in a non-CO$_2$ incubator for 15–20 min at 37°C. Subsequently, the XF Palmitate-BSA FAO Substrates (Agilent) palmitate-BSA (167 μM palmitate conjugated with 28 μM BSA) and 0.5 mM L-carnitine (Sigma Aldrich) were added to the medium for the assessment of OCR. Immediately, XF Cell Mito Stress Test was performed in a Seahorse XFp Analyzer (Agilent) by sequentially adding several modulators of mitochondrial function, including 1 mM oligomycin (Oligo) (Agilent), 1.5 mM fluorocarbonyl cyanide phenylhydrazone (FCCP) (Agilent), and 100 nM rotenone plus 1 mM antimycin A (Rot/Ant A) (Agilent). Data were analyzed with Wave software version 2.3.0 (Agilent) and proper statistical methods using GraphPad Prism version 6 (GraphPad Software).

## $^{13}$C-tracing assessment

Cell samples cultured in a medium containing [U-$^{13}$C] glucose (11 mM) were mixed with 10 μl formate and 800 μl chloroform, and processed by four cycles of 1 min ultra-sonication and 1 min interval in ice-water bath followed by standing for 30 min at room temperature prior to centrifugation at 3000 $\times g$ for 15 min. All chloroform was transferred to a new glass vial and mixed with 500 μl 75% ethanol (containing 0.5 M KOH) prior to incubation at 80°C for 60 min. Then, the mixture was added with 600 μl of hexane and vortexed for 1 min and made to stand for 30 min. After centrifugation at 3000 $\times g$ for 15 min, all hexane layers were evaporated to dryness. The dry residue deposit was mixed with 10 μl 1-hydroxybenzotriazole (HoBt) (in dimethyl sulfoxide [DMSO]), 20 μl cholamine (in DMSO with 200 mM triethylamine [TEA]), and 10 μl O-(7-azabenzotriazol-1yl)-N,N,N',N'-tetramethyluronium hexafluorophosphate (HATU) (in DMSO) followed by incubating for 5 min at room temperature. 60 μl acetonitrile was added, followed by centrifuging at 3000 $\times g$ for 15 min at 4°C prior to UHPLC-HRMS analysis.

The ultra-high performance liquid chromatography-tandem mass-spectrometry (UHPLC-MS/MS) analysis was performed on an Agilent 1290 Infinity II UHPLC system coupled to a 6470A Triple Quadrupole mass spectrometre (Santa Clara, CA, United States). Samples were injected onto a Waters UPLC BEH C18 column (100 mm $\times$ 2.1 mm, 1.7 μm) at a flow rate of 0.3 ml/min. The mobile phase consisted of water (phase A) and acetonitrile (phase B), both with 0.1% formate. The chromatographic separation was conducted by a gradient elution program as follows: 0 min, 10% B; 4 min, 40% B; 8 min, 45% B; 11 min, 50% B; 14 min, 70% B; 15 min, 90% B; 15.5 min, 100% B; 18 min, 100% B; 18.1 min, 10% B; 20 min, 10% B. The column temperature was 40°C. The raw data were processed by Agilent MassHunter Workstation Software (version B.08.00) by using the default parameters and assisting manual inspection to ensure the qualitative and quantitative accuracies of each compound. The peak areas of target compounds were integrated and the output was run for IsoCorrection analysis.

## Assessment of metabolic enzyme activities

Cells were lysed by ultrasonic treatment in extracting buffer from the kits. Supernatants were collected and used for the analyses of enzymatic activities. ACC Activity Assay Kit (Suzhou Comin), ACLY Activity Assay Kit (Suzhou Comin), FASN Activity Assay Kit (Suzhou Comin), and CPT1 Activity Assay Kit (Suzhou Comin) were used according to the manufacturers' instructions.

## Immunoprecipitation and western blotting

Cells were lysed in cold western blot-immunoprecipitation (WB-IP) lysis buffer (Beyotime) for 30 min and centrifuged (at 4°C, 5 min, at 14,000 $\times g$) to remove cell debris. The supernatant was collected as a whole-cell lysate. ACLY, ubiquitin (Ub), and CUL3 were immunoprecipitated from the whole-cell lysates using anti-ACLY (Abcam), anti-ubiquitin (Abcam), and anti-CUL3 (Santa Cruz Biotechnology) antibodies coupling to Protein A/G PLUS Agarose Beads (Santa Cruz Biotechnology) overnight at 4°C. Nonspecific rabbit or mouse IgG antibody was used as a negative control. Immunoprecipitated proteins were mixed with 1$\times$ loading buffer and boiled at 100°C for 10 min followed by WB. For

WB, protein samples were resolved by sodium dodecyl sulfate-polyacrylamide gel electrophoresis gels and then proteins were transferred to polyvinylidene fluoride membranes (Millipore). The membranes were blocked with 5% nonfat milk for 1 hr at room temperature before incubation with primary antibodies overnight at 4°C. Mouse- or rabbit-conjugated horseradish peroxidase (HRP) secondary antibodies were added to the membranes for 50 min at room temperature, followed by washing three times with 1× Tris-buffered saline with Tween 20 (TBST), and visualized using Chemiluminescent HRP Substrate (Millipore) on Chemiluminescence Image System (Tanon Science and Technology Co, Ltd). The antibodies for WB are shown in Key Resources Table.

### In vitro iTreg suppression assay

Naive CD4$^+$ T cells from WT or $Cd4^{Cre}Cul3^{fl/fl}$ mice were transfected with $Acly$-specific or scramble siRNA and cultured under Th0- or iTreg-polarization condition for 72 hr to obtain Th0 (WT), iTreg (WT), iTreg ($Cul3^{-/-}$), and iTreg ($Cul3^{-/-}$+si$Acly$) cells. Meanwhile, $2 \times 10^5$ freshly isolated naive CD4$^+$ T cells from WT mice pulse-labeled with carboxyfluorescein diacetate succinimidyl ester (CFSE) (Biolegend) at 0.5 μM for 20 min were co-cultured with different groups of iTreg cells at a ratio of 1:1 in the presence of Dynabeads Mouse T-Activator CD3/CD28 for 72 hr as described previously (*De Rosa et al., 2015*). The flow cytometry was used for the assessment of CFSE dilution, which indicates the suppressive function of iTreg cells in vitro.

Naive CD4$^+$ T cells from $Foxp3^{YFP-Cre}$ mice were transfected with siRNAs against $Acly$ and (or) $Cul3$ and cultured under Th0- or iTreg-polarization condition for 72 hr followed by sorting yellow fluorescent protein (YFP)-positive cells to obtain Th0, YFP-iTreg, YFP-iTreg (si$Cul3$), and YFP-iTreg (si$Cul3$+si$Acly$) cells. Meanwhile, $2 \times 10^5$ freshly isolated naive CD4$^+$ T cells from WT mice pulse-labeled with CellTrace Violet (CTV) (Thermo Fisher) at 1 μM for 20 min were co-cultured with different groups of iTreg cells at a ratio of 1:1 in the presence of Dynabeads Mouse T-Activator CD3/CD28 for 72 hr as described previously. The flow cytometry was used for the assessment of CTV dilution (*Johnson et al., 2018*), which indicates the suppressive function of iTreg cells in vitro.

### Mouse model of colitis

T-cell transfer colitis based on adoptive transfer of naive CD4$^+$ T cells into $Rag1^{-/-}$ mice is widely used for studying chronic intestinal inflammation (*Furusawa et al., 2013*). As described previously, $5 \times 10^5$ freshly isolated naive CD4$^+$ T cells were administrated into $Rag1^{-/-}$ mice by intraperitoneal injection to induce colitis. Concomitant injection of $5 \times 10^5$ Th0 (WT), iTreg (WT), iTreg ($Cul3^{-/-}$), or iTreg ($Cul3^{-/-}$+si$Acly$) cells was conducted for the evaluation of CUL3 and ACLY function. Meanwhile, $Rag1^{-/-}$ mice of blank control group were administrated with PBS. Moreover, the function of ACLY in acute colitis was also assessed in DSS-induced colitis model. DSS (MP Biomedicals) dissolved in drinking water (2%, w/v) was given ad libitum to 6- to 8-week-old female C57BL/6J mice for 7 days followed by water for 3 days (*Wirtz et al., 2017*; *Wirtz et al., 2007*). Mice supplied with regular drinking water were included as a negative control. ACLY inhibitor, SB204990 (37.5 mg/kg/d) (Tocris Bioscience), was administered orally every 2 days before the assessment of colitis.

### Evaluation of colitis in mice

To assess the severity of colitis, DAI was weekly (T-cell transfer colitis) or daily (DSS-induced colitis) recorded by scoring the body weight loss, stool consistency, and blood in the stool as described in the literature (*Furusawa et al., 2013*; *Wirtz et al., 2017*; *Wirtz et al., 2007*). 7 weeks later (T-cell transfer colitis) or 10 days later (DSS-induced colitis), mice were euthanized for the examination of histopathological changes in colon tissues. Entire colons were removed and colon length was determined. Following cleaning with saline and fixation with formalin, colon samples were embedded in paraffin. Sections (3 μm) were stained with hematoxylin and eosin (H and E) and slides were blindly analyzed for intestinal inflammation according to the previous description (*Wirtz et al., 2017*; *Wirtz et al., 2007*). In addition, for DSS-induced colitis, the mononuclear cells from MLNs and the cLP were prepared as described previously (*Atarashi et al., 2011*) for the analysis of Treg cells by flow cytometry.

## Induction of allergic diarrhea

Mice were sensitized with 50 µg of OVA (Sigma Aldrich) and 1 mg of alum (Thermo) in sterile saline by interperitoneal (ip) injection. From day 14 on, mice were challenged by oral gavage with 50 mg of OVA in saline three times per week for 3 weeks as described. Mice showing profuse liquid stool within 60 min after oral OVA challenge were recorded as diarrhea-positive. Th0 and iTreg cells under various conditions were adoptively transferred into mice, respectively, by tail vein injection (iv) after the first and forth challenge. 2 hr after the last oral administration, mice were sacrificed and blood samples and intestinal tissues were collected for detection of OVA-specific IgE and MCPT1 in serum by ELISA and eosnophil infiltration by H and E staining.

## Data analysis and statistics

All experiments were done at least three times independently. Statistical analysis was performed using GraphPad Prism version 6 (GraphPad Software). Data are displayed as mean ± standard (SD). The p-values were calculated from Student's unpaired t-test when comparing within two groups. One-way or two-way analysis of variance (ANOVA) was performed in the indicated figures when more than two groups were compared. p-values <0.05 were considered significant. p-values were indicated on graphs as *$p<0.05$, **$p<0.01$, ***$p<0.001$, ****$p<0.0001$; ns, nonsignificant.

## Acknowledgements

We thank Jeffrey D Singer (Department of Biology, Portland State University, Portland, OR, USA) and Zichun Hua (The State Key Laboratory of Pharmaceutical Biotechnology, Department of Biochemistry, College of Life Sciences, Nanjing University, Nanjing) for generously sponsoring the $Cul3^{fl/fl}$ and $Cd4^{Cre}$ mice, respectively. We also thank Weiwei Yang (Shanghai Institute of Biochemistry and Cell Biology, Chinese Academy of Sciences, China) for critical comments. This work was supported by grants from the National Natural Science Foundation of China (32070896 and 31870896) and the Fundamental Research Funds for the Central Universities (2412019FZ028).

## Additional information

### Funding

| Funder | Grant reference number | Author |
|---|---|---|
| National Natural Science Foundation of China | 32070896 | Min Wei |
| National Natural Science Foundation of China | 31870896 | Min Wei |
| Fundamental Research Funds for the Central Universities | 2412019FZ028 | Yunpeng Feng |

The funders had no role in study design, data collection and interpretation, or the decision to submit the work for publication.

### Author contributions

Miaomiao Tian, Fengqi Hao, Conceptualization, Data curation, Software, Formal analysis, Validation, Visualization, Methodology, Writing - original draft, Writing - review and editing; Xin Jin, Software, Formal analysis, Methodology, Writing - original draft, Writing - review and editing; Xue Sun, Ying Jiang, Data curation, Formal analysis, Visualization, Methodology; Yang Wang, Formal analysis, Methodology; Dan Li, Resources; Tianyi Chang, Pinghui Peng, Yuanxi Li, Data curation; Yingying Zou, Software, Methodology; Chaoyi Xia, Jia Liu, Formal analysis; Ping Wang, Conceptualization, Resources, Formal analysis; Yunpeng Feng, Conceptualization, Supervision, Funding acquisition, Writing - original draft, Writing - review and editing; Min Wei, Conceptualization, Supervision, Funding acquisition, Validation, Writing - original draft, Project administration, Writing - review and editing

## Author ORCIDs

Fengqi Hao https://orcid.org/0000-0002-8640-7244
Yunpeng Feng https://orcid.org/0000-0002-6912-3338
Min Wei https://orcid.org/0000-0001-9945-6961

## Ethics

Human subjects: All work with human blood samples was approved by the local ethics committee (Reference number: AP2019085, the ethical committee of the Northeast Normal University), and informed consent was obtained from all subjects.

Animal experimentation: All mouse experiments were conducted in accordance with the protocols for animal use, treatment, and euthanasia approved by the institutional animal care and use committee (IACUC) of Northeast Normal University (Reference number: AP2019085). The number of experimental mice used for research purposes is reduced as far as possible while still allowing the scientific purpose of the research to be achieved.

## Decision letter and Author response

Decision letter https://doi.org/10.7554/eLife.62394.sa1
Author response https://doi.org/10.7554/eLife.62394.sa2

# Additional files

## Supplementary files

• Source data 1. Original data and blots supporting main figures and figure supplements.

• Supplementary file 1. List of oligonucleotides. Primers for PCR and quantitative PCR, and sequences of siRNAs are included.

• Transparent reporting form

## Data availability

All data generated or analysed during this study are included in the manuscript and supporting file; Source Data files have been provided for Figures 1-6.

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
