## [Decision Letter]

**Acceptance summary:**

In this study, the authors reveal downregulation of ACLY expression as a metabolic determinant of iTreg cell generation induced by TGFβ1. The authors show that neutral lipid content is decreased during iTreg compared to Th0 differentiation. Using complementary systems of overexpression, gene repression, and drug inhibition, the authors established that ACLY expression or activity is correlated with alterations in neutral lipid content in iTreg cells, and more importantly, changes in iTreg differentiation in both mouse and human cells. Mechanistically, the authors show that ACLY is targeted for ubiquitination and proteasomal degradation by CUL3-KLHL25 and further demonstrate that the CUL3-ACLY axis is important for regulating iTreg cell-dependent suppression of T cell proliferation and development of colitis and allergic diarrhea in vivo. This study will be of interest to those in the immunometabolism field and will have broader interest to those studying the therapeutic efficacy of ACLY inhibition in certain diseases.

**Decision letter after peer review:**

Thank you for submitting your article "ACLY ubiquitination by CUL3-KLHL25 coordinates fatty acid synthesis to promote iTreg differentiation" for consideration by *eLife*. Your article has been reviewed by 2 peer reviewers, one of whom is a member of our Board of Reviewing Editors, and the evaluation has been overseen by Tadatsugu Taniguchi as the Senior Editor. The reviewers have opted to remain anonymous.

The reviewers have discussed the reviews with one another and the Reviewing Editor has drafted this decision to help you prepare a revised submission.

Summary:

In this study, the authors show that neutral lipid content is decreased during iTreg compared to Th0 differentiation. Using complementary systems of overexpression, gene repression, and drug inhibition, the authors established that ACLY expression or activity is correlated with alterations in neutral lipid content in iTreg cells, and more importantly, changes in iTreg differentiation in both mouse and human cells. Mechanistically, the authors show that ACLY is targeted for ubiquitination and proteasomal degradation by CUL3-KLHL25 and further demonstrate that the CUL3-ACLY axis is important for regulating iTreg cell-dependent suppression of T cell proliferation and development of colitis in vivo.

This is a well performed study that reveals downregulation of ACLY expression as a metabolic determinant of iTreg cell differentiation induced by TGFβ1.

Overall, the experiments were well performed and the study clearly shows the important roles of CUL3-KLHL25-mediated ACLY ubiquitination in iTreg differentiation. We have the following suggestions to improve the manuscript.

Essential revisions:

1. The authors provide convincing, but correlative, data for decreased neutral lipid staining (using BODIPY) in iTreg cells that is impacted by altered expression or activity of ACLY. BODIPY staining alone cannot distinguish if these changes are due only to alterations of de novo fatty acid synthesis. Indeed, such differences may also reflect changes in lipid uptake or even degradation via the fatty acid oxidation pathway. In addition, ACLY-derived acetyl-CoA can also serve as a precursor for the mevalonate-cholesterol biosynthesis pathway, as well as a substrate for acetyltransferase enzymes. Additional experiments are therefore required to demonstrate that downregulation of ACLY-dependent fatty acid synthesis is essential for iTreg cell differentiation, including:

a. Assess de novo lipid synthesis using radiolabeled acetate or glucose in ACLY-sufficient or -deficient iTreg (or SB204990-treated) cells to establish that the change in BODIPY staining is attributed to reduced lipid synthesis.

b. Add back palmitate, mevalonate, cholesterol, or acetate to iTreg cell cultures to establish a more direct relationship between ACLY downregulation/inhibition and fatty acid synthesis for iTreg cell generation.

c. Assess changes in histone acetylation by immunoblot, imaging, or flow cytometry analysis.

d. Measure nuclear levels of acetyl-CoA, as ACLY has been reported to enter the nucleus and influence histone acetylation (Sivanand et al., 2017 Mol Cell). The authors currently only assess changes in non-nuclear acetyl-CoA levels.

2. The authors suggest that downregulation of ACLY is important for promoting activation of CPT1a-dependent fatty acid oxidation, which has been implicated for iTreg cell generation in vitro. Do ACLY-deficient or SB204990-treated iTreg cells have increased rates of fatty acid oxidation?

3. TGFβ1 is also important for the development of thymic-derived Treg cells (Liu et al. 2008 Nat Immunol; Ouyang et al. 2010 Immunity). Is ACLY expression altered as cells differentiate from double-positive thymocytes to Treg precursors to mature Treg cells in the thymus?

4. For data presented in Figure 5, it remains unclear if the altered suppressive activity of CUL3- +/- ACLY-deficient iTreg cells in vitro or in vivo is due to differences in the proportion of iTreg cells or changes in suppressive function. To address this, the authors should perform the below experiments:

a. Assess the expression of surrogate molecules associated with Treg cell suppressive activity on Foxp3^+^ iTreg cells under the various conditions (e.g. CTLA4, ICOS, TIGIT, IL-10, etc.).

b. Assess the number of Treg cells in the intestine of *Cd4*-cre:*Cul3*-deficient mice. Also, the authors should co-stain the intestinal Treg cells with RORγt and Helios to better distinguish peripherally-induced and thymic-derived Treg cells in the intestines (Sefik et al., 2015 Science; Ohnmacht et al., 2015 Science).

c. Use naïve T cells from Foxp3 reporter mice and sort purify Foxp3^+^ iTreg cells for the in vitro suppressive assay.

d. Would be interesting if the authors investigate, for ex, allergic diarrhea, in which intestinal Treg cells are implicated in the prevention.

5. The therapeutic relevance of SB204990 in the DSS colitis model shown in Figure S8 should be discussed in the Results section of the manuscript, not only the Discussion.

---

## [Author Response]

Essential revisions:1. The authors provide convincing, but correlative, data for decreased neutral lipid staining (using BODIPY) in iTreg cells that is impacted by altered expression or activity of ACLY. BODIPY staining alone cannot distinguish if these changes are due only to alterations of de novo fatty acid synthesis. Indeed, such differences may also reflect changes in lipid uptake or even degradation via the fatty acid oxidation pathway. In addition, ACLY-derived acetyl-CoA can also serve as a precursor for the mevalonate-cholesterol biosynthesis pathway, as well as a substrate for acetyltransferase enzymes. Additional experiments are therefore required to demonstrate that downregulation of ACLY-dependent fatty acid synthesis is essential for iTreg cell differentiation, including:

We totally agree with reviewers that comprehensive characterization of ACLY-repression-induced alterations in cell metabolism during iTreg cell differentiation is critical and pivotal. Thus we have performed a series of new experiments including radio-labelling and rescue assays as suggested to improve this manuscript.

a. Assess de novo lipid synthesis using radiolabeled acetate or glucose in ACLY-sufficient or -deficient iTreg (or SB204990-treated) cells to establish that the change in BODIPY staining is attributed to reduced lipid synthesis.

As the reviewer pointed out, BODIPY staining alone cannot pinpoint the contribution from ACLY-dependent alteration in FAS to de novo lipid synthesis. We thus have tracked [U-^13^C] glucose in iTreg cells inhibited for ACLY by SB204990 and found moderate to significant reductions in different ^13^C-labelled fatty acids (new Figure 2A). This is consistent with decreased BODIPY staining in response to ACLY inhibition (previous Figure 1H) and again supports an important role for ACLY in de novo lipid synthesis. In addition, we have explained the isotope tracing analysis of FAS in the revised manuscript (please see page 6 line 135-138).

b. Add back palmitate, mevalonate, cholesterol, or acetate to iTreg cell cultures to establish a more direct relationship between ACLY downregulation/inhibition and fatty acid synthesis for iTreg cell generation.

Following the reviewer’s constructive suggestion, we have performed a series of rescue assays to carefully evaluate the functional consequences of ACLY-dependent metabolic alterations in iTreg differentiation. Upon the addition of mevalonate, cholesterol or acetate, upregulated iTreg differentiation induced by ACLY inhibition remained unchanged (new Figure 2—figure supplement 2A-C, Figure R1A-B). In line with this observation, levels of HMG-CoA, mevalonate, mevalonate-5-pyrophosphate, cholesterol as well as nuclear acetyl-CoA were unaffected in response to ACLY inhibition (new Figure 2—figure supplement 1A-D, Figure 2—figure supplement 3A-B). In addition, histone acetylation remained rather stable regardless of whether ACLY was inhibited or not (new Figure 2—figure supplement 3C-D). These results indicate that ACLY are unlikely to regulate iTreg differentiation via mevalonate-cholesterol biosynthesis pathway or histone acetylation.

Despite a reduction in FAS resulted from ACLY inhibition (new Figure 2A), addback of palmitate still could not rescue changes in iTreg differentiation induced by ACLY inhibition (new Figure 2—figure supplement 4A, Figure 2B). One may consider that the supplement of a single end-product is probably not sufficient enough to rescue the whole FAS. To evaluate the role of individual intermediate metabolite from FAS in ACLY-dependent regulation of iTreg differentiation, we separately blocked ACC and FASN in cells inhibited for ACLY by SB204990 (new Figure 2—figure supplement 4B). FASN inhibition by cerulenin, but not ACC inhibition by PF05175157, effectively abrogated ACLY-inhibition-induced iTreg differentiation (new Figure 2—figure supplement 4C), suggesting an important role for malonyl-CoA in ACLY-dependent regulation of iTreg differentiation. As the substrate of FASN, malonyl-CoA declined upon ACLY inhibition (new Figure 2—figure supplement 4D). Importantly, adding malonyl-CoA back totally blocked ACLY-inhibition-induced changes in iTreg differentiation (new Figure 2C). Together, these data argue that ACLY inhibition promotes iTreg differentiation probably by reducing malonyl-CoA. In the revised manuscript, these data have been selectively integrated and the main text has been modified accordingly (please see page 6-7 line 138-165).

c. Assess changes in histone acetylation by immunoblot, imaging, or flow cytometry analysis.

As suggested, we have assayed histone acetylation, which remained largely unchanged irrespective of the presence or absence of ACLY inhibitor (new Figure 2—figure supplement 3C-D). We also compared histone acetylation levels in Th0 and iTreg cells and found they were rather similar (Figure R2A-B). These results indicate that ACLY-dependent regulation of iTreg differentiation may not through histone acetylation. In the revised manuscript, this point has been integrated (please see page 7 line 147-149).

d. Measure nuclear levels of acetyl-CoA, as ACLY has been reported to enter the nucleus and influence histone acetylation (Sivanand et al., 2017 Mol Cell). The authors currently only assess changes in non-nuclear acetyl-CoA levels.

Following the reviewer’s suggestion, we have assessed nuclear acetyl-CoA levels at different conditions. Upon ACLY inhibition, the level of acetyl-CoA in nuclear fraction failed to change significantly (new Figure 2—figure supplement 3A-B). In a parallel assay comparing Th0 to iTreg cells, levels of nuclear acetyl-CoA also seemed rather similar (Figure R2C-D). In line with unchanged histone acetylation upon ACLY inhibition (new Figure 2—figure supplement 3C-D), these results suggest a dispensable role for ACLY in the regulation of nuclear acetyl-CoA level as well as histone acetylation during iTreg differentiation. In addition, we have also discussed the assessment of nuclear acetyl-CoA level in the revised manuscript (please see page 6-7 line 144-147).

2. The authors suggest that downregulation of ACLY is important for promoting activation of CPT1a-dependent fatty acid oxidation, which has been implicated for iTreg cell generation in vitro. Do ACLY-deficient or SB204990-treated iTreg cells have increased rates of fatty acid oxidation?

To address this point, we examined palmitate-dependent oxidation and CPT1 activity, which have been widely used for evaluating the rate of FAO (Hao et al., 2021; Kim et al., 2002). Upon ACLY inhibition, both palmitate-dependent oxidation and CPT1 activity were significantly elevated (new Figure 2D-E, R3A-B). In the revised manuscript, these data have been selectively integrated and the main text has been modified accordingly (please see page 7 line 169-170).

3. TGFβ1 is also important for the development of thymic-derived Treg cells (Liu et al. 2008 Nat Immunol; Ouyang et al. 2010 Immunity). Is ACLY expression altered as cells differentiate from double-positive thymocytes to Treg precursors to mature Treg cells in the thymus?

As suggested, we examined ACLY expression as cells differentiate from double-positive thymocytes to Treg precursors and mature Treg cells. During thymic-derived Treg cell development, ACLY expression increased gradually (Figure R5). Levels of ACLY in Treg precursors (CD4^+^CD8^-^CD25^+^Foxp3^-^) and mature Treg cells (CD4^+^CD8^-^CD25^+^Foxp3^+^) are significantly higher than that in double-positive thymocytes (CD4^+^CD8^+^) (Figure R5).

In contrast to iTreg cells that are dependent on FAO, tTreg cells have been found to rely on glycolysis and lipid synthesis (Priyadharshini et al., 2018; Zeng et al., 2013). Although both iTreg and tTreg differentiation are reliant on TGFβ1, different levels of ACLY may be of help for cells to establish distinct metabolic patterns. For instance, relatively high level of ACLY in tTreg cells (Figure R5) can support efficient lipid synthesis, which is known to be important for the suppressive function of tTreg cells (Zeng et al., 2013).

4. For data presented in Figure 5, it remains unclear if the altered suppressive activity of CUL3- +/- ACLY-deficient iTreg cells in vitro or in vivo is due to differences in the proportion of iTreg cells or changes in suppressive function. To address this, the authors should perform the below experiments:a. Assess the expression of surrogate molecules associated with Treg cell suppressive activity on Foxp3^+^ iTreg cells under the various conditions (e.g. CTLA4, ICOS, TIGIT, IL-10, etc.).

We truly appreciate the reviewer’s suggestion and have assessed the expression of surrogate molecules associated with the suppressive activity of iTreg cells under various conditions. Neither CUL3 deficiency (*Cul3^-/-^*) nor the lack of CUL3 and ACLY (*Cul3^-/-^* and si*Acly*) changed the expression of representative surrogate molecules, such as CTLA4, ICOS, TIGIT and IL-10, in Foxp3^+^ iTreg cells (new Figure 4—figure supplement 4A). These data suggest that ACLY-dependent regulation of iTreg is probably not through modulation of its suppressive function. In the revised manuscript, we have integrated these new data and modified the main text accordingly (please see page 10 line 256-259).

b. Assess the number of Treg cells in the intestine of *Cd4*-cre:*Cul3*-deficient mice. Also, the authors should co-stain the intestinal Treg cells with RORγt and Helios to better distinguish peripherally-induced and thymic-derived Treg cells in the intestines (Sefik et al., 2015 Science; Ohnmacht et al., 2015 Science).

As the reviewer suggested, we have assessed the number of CD4^+^Foxp3^+^ Treg cells in intestines from *Cd4^Cre^Cul3^fl/fl^* mice and found a dramatic decrease as compared to WT mice (new Figure 4—figure supplement 3C, Figure R6). Furthermore, both peripherally-induced (Foxp3^+^RORγt^+^) and thymic-derived (Foxp3^+^Helios^+^) Treg cells were reduced (Figure R6). Together, these data confirmed the CUL3-deletion-induced impairment in iTreg differentiation in vivo. In the revised manuscript, these data have been selectively integrated and the main text has been modified accordingly (please see page 10 line 244-246).

c. Use naïve T cells from Foxp3 reporter mice and sort purify Foxp3^+^ iTreg cells for the in vitro suppressive assay.

We took the advantage of the established *Foxp3^YFP-Cre^* mice (Rubtsov et al., 2008) and isolated naïve CD4^+^ T cells for functional assays. Following the transfection of siRNAs against *Cul3* and (or) *Acly*, naïve CD4^+^ T cells were induced under iTreg differentiation condition prior to the analysis of Foxp3-YFP^+^ iTreg cells. Clearly, *Cul3* knockdown by siRNA caused a reduction in Foxp3-YFP^+^ iTreg cell number, which could be blocked by a simultaneous depletion of ACLY (new Figure 4—figure supplement 4B). These observations completely recapitulated our previous in vitro findings (new Figure 4E) and confirmed the important role of CUL3-ACLY axis during iTreg differentiation. In addition, the lack of CUL3 and (or) ACLY did not seem to affect the suppressive function of Foxp3-YFP^+^ iTreg cells, because T cell proliferation remained unchanged, regardless of whether wild-type, si*Cul3*- or si*Cul3*/si*Acly*-treated Foxp3-YFP^+^ iTreg cells were presented in the co-culture system (new Figure 4—figure supplement 4B). Meanwhile, the main text of the revised manuscript has been modified accordingly (please see page 10 line 259-264).

d. Would be interesting if the authors investigate, for ex, allergic diarrhea, in which intestinal Treg cells are implicated in the prevention.

Following the reviewer’s suggestion, we established an OVA-induced allergic diarrhea mouse model for the functional evaluation of CUL3-dependent ACLY degradation in iTreg differentiation (new Figure 6—figure supplement 4A). Upon OVA stimulation, mice developed diarrhea and exhibited upregulated OVA-specific IgE antibodies and mastocyte protease (MCPT-1) in the serum as expected (Kordowski et al., 2019) (new Figure 6—figure supplement 4B-E). Importantly, the adoptive transfer of wild-type (WT) iTreg cells, but not CUL3 deficient (*Cul3^-/-^*) cells, into recipient mice remarkably alleviated diarrhea-associated defects (new Figure 6—figure supplement 4B-E). The alleviation effect from CUL3 deficiency on the diarrhea was rescued, when we used targeting siRNA to knock down *Acly* (new Figure 6—figure supplement 4B-E). Together, these in vivo data argue an important role for CUL3-mediated ACLY ubiquitination in the regulation of diarrhea. In the revised manuscript, we have explained the functional assessment for Treg cells derived from CUL3 and (or) ACLY deficiency in allergic diarrhea (please see page 12 line 311-320).

5. The therapeutic relevance of SB204990 in the DSS colitis model shown in Figure S8 should be discussed in the Results section of the manuscript, not only the Discussion.

Following the reviewer’s suggestion, we have discussed the therapeutic relevance of SB204990 in the DSS colitis model in the Results section of the manuscript (Page 12 line 304-307).

References

Essig K, Hu D, Guimaraes JC, Alterauge D, Edelmann S, Raj T, Kranich J, Behrens G, Heiseke A, Floess S, Klein J, Maiser A, Marschall S, Hrabĕ de Angelis M, Leonhardt H, Calkhoven CF, Noessner E, Brocker T, Huehn J, Krug AB, Zavolan M, Baumjohann D, Heissmeyer V. 2017. Roquin suppresses the PI3K-mTOR signaling pathway to inhibit T helper cell differentiation and conversion of Treg to Tfr cells. Immunity 47:1067-1082.e12. doi:10.1016/j.immuni.2017.11.008

Hao F, Tian M, Zhang X, Jin X, Jiang Y, Sun X, Wang Y, Peng P, Liu J, Xia C, Feng Y, Wei M. 2021. Butyrate enhances CPT1A activity to promote fatty acid oxidation and iTreg differentiation. Proc Natl Acad Sci 118:e2014681118. doi:10.1073/pnas.2014681118

Huang Q, Xu L, Ye L. 2019. T cell immune response within B-cell follicles. Advances in Immunology in China – Part A. pp. 155–171. doi:10.1016/bs.ai.2019.08.008

Kim J-Y, Koves TR, Yu G-S, Gulick T, Cortright RN, Dohm GL, Muoio DM. 2002. Evidence of a malonyl-CoA-insensitive carnitine palmitoyltransferase I activity in red skeletal muscle. Am J Physiol Metab 282:E1014–E1022. doi:10.1152/ajpendo.00233.2001

Kordowski A, Reinicke AT, Wu D, Orinska Z, Hagemann P, Huber-Lang M, Lee JB, Wang YH, Hogan SP, Köhl J. 2019. C5a receptor 1−/− mice are protected from the development of IgE-mediated experimental food allergy. Allergy Eur J Allergy Clin Immunol 74:767–779. doi:10.1111/all.13637

MacIver NJ, Michalek RD, Rathmell JC. 2013. Metabolic regulation of T lymphocytes. Annu Rev Immunol 31:259–283. doi:10.1146/annurev-immunol-032712-095956

Mathew R, Seiler MP, Scanlon ST, Mao A, Constantinides MG, Bertozzi-Villa C, Singer JD, Bendelac A. 2012. BTB-ZF factors recruit the E3 ligase cullin 3 to regulate lymphoid effector programs. Nature 491:618–621. doi:10.1038/nature11548

Michalek RD, Gerriets VA, Jacobs SR, Macintyre AN, MacIver NJ, Mason EF, Sullivan SA, Nichols AG, Rathmell JC. 2011. Cutting edge: Distinct glycolytic and lipid oxidative metabolic programs are essential for effector and regulatory CD4+ T cell subsets. J Immunol 186:3299–3303. doi:10.4049/jimmunol.1003613

O’Connor RS, Guo L, Ghassemi S, Snyder NW, Worth AJ, Weng L, Kam Y, Philipson B, Trefely S, Nunez-Cruz S, Blair IA, June CH, Milone MC. 2018. The CPT1a inhibitor, etomoxir induces severe oxidative stress at commonly used concentrations. Sci Rep 8:1–9. doi:10.1038/s41598-018-24676-6

Patel CH, Powell JD. 2017. Targeting T cell metabolism to regulate T cell activation, differentiation and function in disease. Curr Opin Immunol 46:82–88. doi:10.1016/j.coi.2017.04.006

Priyadharshini B, Loschi M, Newton RH, Zhang J, Finn KK, Gerriets VA, Huynh A, Rathmell JC, Blazar BR, Turka LA. 2018. Cutting edge: TGF-β and phosphatidylinositol 3-kinase signals modulate distinct metabolism of regulatory T cell subsets. J Immunol 201:2215–2219. doi:10.4049/jimmunol.1800311

Raud B, Roy DG, Divakaruni AS, Tarasenko TN, Franke R, Ma EH, Samborska B, Hsieh WY, Wong AH, Stüve P, Arnold-Schrauf C, Guderian M, Lochner M, Rampertaap S, Romito K, Monsale J, Brönstrup M, Bensinger SJ, Murphy AN, McGuire PJ, Jones RG, Sparwasser T, Berod L. 2018. Etomoxir actions on regulatory and memory T cells are independent of Cpt1a-mediated fatty acid oxidation. Cell Metab 28:504-515.e7. doi:10.1016/j.cmet.2018.06.002

Rubtsov YP, Rasmussen JP, Chi EY, Fontenot J, Castelli L, Ye X, Treuting P, Siewe L, Roers A, Henderson WR, Muller W, Rudensky AY. 2008. Regulatory T cell-derived interleukin-10 limits inflammation at environmental interfaces. Immunity 28:546–558. doi:10.1016/j.immuni.2008.02.017

Saravia J, Zeng H, Dhungana Y, Bastardo Blanco D, Nguyen T-LM, Chapman NM, Wang Y, Kanneganti A, Liu S, Raynor JL, Vogel P, Neale G, Carmeliet P, Chi H. 2020. Homeostasis and transitional activation of regulatory T cells require c-Myc. Sci Adv 6:eaaw6443. doi:10.1126/sciadv.aaw6443

Zeng H, Yang K, Cloer C, Neale G, Vogel P, Chi H. 2013. mTORC1 couples immune signals and metabolic programming to establish Treg-cell function. Nature 499:485–490. doi:10.1038/nature12297

Zhang D, Chia C, Jiao X, Jin W, Kasagi S, Wu R, Konkel JE, Nakatsukasa H, Zanvit P, Goldberg N, Chen Q, Sun L, Chen Z, Chen W. 2017. D-mannose induces regulatory T cells and suppresses immunopathology. Nat Med 1–12. doi:10.1038/nm.4375